# On Inductive Biases That Enable Generalization of Diffusion Transformers

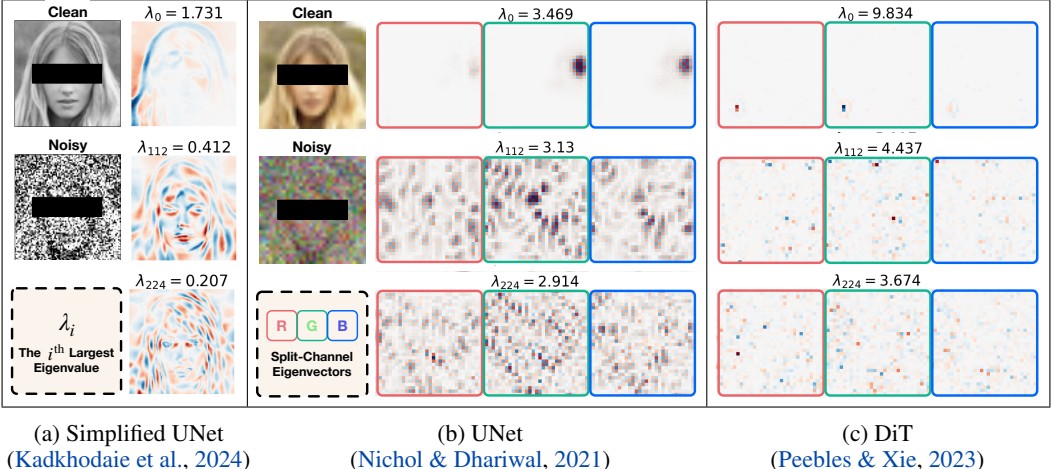

| (a) Simplified UNet | (b) UNet | (c) DiT |
|---|---|---|
| (Kadkhodaie et al., 2024) | (Nichol & Dhariwal, 2021) | (Peebles & Xie, 2023) |

Figure 1: Jacobian eigenvectors of (a) a simplified one-channel UNet, (b) the UNet introduced in improved diffusion (Nichol & Dhariwal, 2021), and (c) a DiT (Peebles & Xie, 2023). Kadkhodaie et al. (2024) find that the generalization of a UNet-based diffusion model is driven by geometry-adaptive harmonic bases (a), which display oscillatory patterns whose frequency increases as the eigenvalue $\lambda_k$ decreases. We observe similar harmonic bases in split-channel eigenvectors (b) with standard UNets (Nichol & Dhariwal, 2021). However, a DiT (Peebles & Xie, 2023) does not exhibit such harmonic bases (c), motivating our investigation into alternative inductive bias of a DiT that enables its generalization. The RGB channels of the split-channel eigenvectors are outlined with red, green, blue boxes, respectively. All models operate directly in the pixel space without applying the patchify operation.

## Abstract

Recent work studying the generalization of diffusion models with UNet-based denoisers reveals inductive biases that can be expressed via geometry-adaptive harmonic bases. However, in practice, more recent denoising networks are often based on transformers, *e.g.*, the diffusion transformer (DiT). This raises the question: do transformer-based denoising networks exhibit inductive biases that can also be expressed via geometry-adaptive harmonic bases? To our surprise, we find that this is *not* the case. This discrepancy motivates our search for the inductive bias that can lead to good generalization in DiT models. Investigating a DiT's pivotal attention modules, we find that locality of attention maps are closely associated with generalization. To verify this finding, we modify the generalization of a DiT by restricting its attention windows. We inject local attention windows to a DiT and observe an improvement in generalization. Furthermore, we empirically find that both the placement and the effective attention size of these local attention windows are crucial factors. Experimental results on the CelebA, ImageNet, and LSUN datasets show that strengthening the inductive bias of a DiT can improve both generalization and generation quality when less training data is available. Source code will be released publicly upon paper publication.

## 1 INTRODUCTION

Diffusion models have achieved remarkable success in visual content generation. Their training involves approximating a distribution in a high-dimensional space from a limited number of training samples–a task that is highly challenging due to the curse of dimensionality. Nonetheless, recent diffusion models (Sohl-Dickstein et al., 2015; Song et al., 2020; Ho et al., 2020; Kadkhodaie & Simoncelli, 2020; Nichol & Dhariwal, 2021; Song et al., 2020) learn to generate high-quality images (Nichol et al., 2021; Dhariwal & Nichol, 2021; Saharia et al., 2022; Rombach et al., 2022; Chen et al., 2023; 2024a) and even videos (Singer et al., 2022; Ho et al., 2022; Girdhar et al., 2023; Blattmann et al., 2023; OpenAI, 2024) using relatively few samples when compared to the underlying high-dimensional space. This indicates that diffusion models exhibit powerful inductive biases (Wilson & Izmailov, 2020; Goyal & Bengio, 2022; Griffiths et al., 2024) that promote effective generalization. What exactly are these powerful inductive biases? Understanding them is crucial for gaining deeper insights into the behavior of diffusion models and their remarkable generalization.

Recent work by Kadkhodaie et al. (2024) on UNet-based diffusion models reveals that the strong generalization of UNet-based denoisers is driven by inductive biases that can be expressed via a set of geometry-adaptive harmonic bases (Mallat et al., 2020). Their result is illustrated in Fig. 1 (a): the harmonic bases are extracted from a simplified one-channel UNet via the eigenvectors of the denoiser's Jacobian matrix. It is easy to extend the analysis of Kadkhodaie et al. (2024) to show that similar harmonic bases are also observed in more complex and classic multi-channel UNets (Nichol & Dhariwal, 2021), as shown in Fig. 1 (b). Given this observation, it is natural to ask: does the emergence of harmonic bases also occur in compelling recent transformer-based diffusion model backbones, *e.g.*, diffusion transformers (DiTs) (Peebles & Xie, 2023)? To explore this possibility, we perform an eigendecomposition of a DiT's Jacobian matrix, following Kadkhodaie et al. (2024). To our surprise, as shown in Fig. 1 (c), a DiT trained in the pixel space does *not* exhibit geometry-adaptive harmonic bases, making it different from a UNet. Building on these insights, a natural question arises: what are the inductive biases that enable the strong generalization of DiTs?

Answering this question is particularly important because of the growing adoption of DiTs in recent methods (Chen et al., 2024b; Esser et al., 2024), partly for its observed performance at scale (Peebles & Xie, 2023). In a new study in this paper, using the PSNR gap (Kadkhodaie et al., 2024) as a metric to evaluate the generalization of diffusion models, we confirm that a DiT indeed exhibits better generalization than a UNet with the same FLOPs. Yet, as mentioned before, this observation alone doesn't reveal the inductive biases which enable generalization.

The generalization mechanism of a DiT may differ from that of UNet-based models, potentially due to the self-attention (Vaswani, 2017) dynamics which are pivotal in DiT models but not in UNets. In a self-attention layer, the attention map, derived from the multiplication of query and key matrices, determines how the value matrix obtained from input tensors influences output tensors. To shed some light, we analyze the attention maps of a DiT and show that locality of the attention maps is closely tied to its generalization ability. Specifically, the attention maps of a DiT trained with insufficient images, *i.e.*, with weak generalization, exhibit a more position-invariant pattern: the output tokens of a self-attention layer are largely influenced by a certain combination of input tensors, irrespective of their positions. In contrast, the attention maps of a DiT trained with sufficient images, which demonstrates strong generalization, exhibit a sparse diagonal pattern. This indicates that each output token is primarily influenced by its neighboring input tokens. This analysis provides insight into how the generalization ability of DiTs can be modified, if necessary, such as when only a small number of training images are available.

Restricting the attention window in self-attention layers should permit modifying a DiT's generalization. Indeed, we find that employing local attention windows (Beltagy et al., 2020; Hassani et al., 2023) is effective. A local attention window restricts the dependence of an output token on its nearby input tokens, thereby promoting the locality of attention maps. In addition, the placement of attention window restrictions within the DiT architecture and the effective size of attention windows are critical factors to steer a DiT's generalization. Our experiments show that placing attention window restrictions in the early attention layers of the DiT architecture yields the most benefit. Experimental results on the CelebA (Liu et al., 2015), ImageNet (Deng et al., 2009), and LSUN (Yu et al., 2015) (bedroom, church, tower, bridge) datasets demonstrate that applying attention window restrictions improves generalization, as reflected by a reduced PSNR gap. We also observe an improved

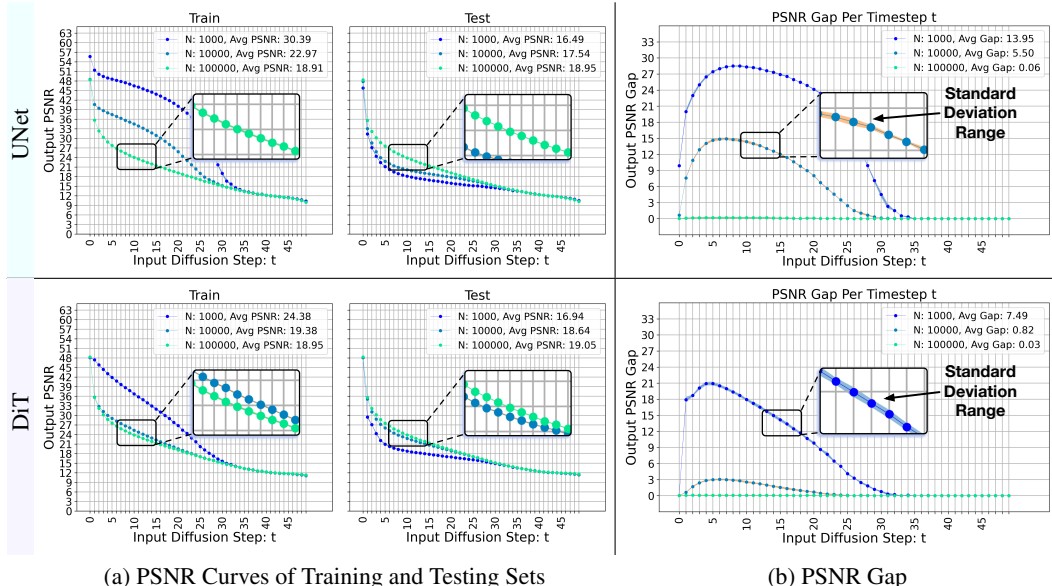

(a) PSNR Curves of Training and Testing Sets     (b) PSNR Gap

Figure 2: The PSNR (a) and PSNR gap (b) comparisons between a UNet and a DiT with the same FLOPs for different training image quantities ($N$). When $N=10^5$, both DiT and UNet show small PSNR gaps between the training and testing sets. Nevertheless, when $N=10^3$ and $N=10^4$, a DiT exhibits smaller PSNR gaps compared to a UNet, indicating a better generalization ability under insufficient training data. All PSNR and PSNR gap curves are averaged over three models trained on different dataset shuffles. The standard deviations, illustrated by the curve shadows in the zoomed-in windows, are negligible, indicating minimal variation.

FID (Heusel et al., 2017) when training with insufficient data, confirming that DiT's generalization can be successfully modified through attention window restrictions.

In summary, the contributions of this paper include the following: 1) We identify the locality of attention maps as a key inductive bias contributing to the generalization of a DiT, and 2) we demonstrate how to control this inductive bias by incorporating local attention windows into a DiT. Enhancing the locality in attention computations effectively modifies a DiT's generalization, resulting in a lower PSNR gap and improved FID scores when insufficient training images are available.

## 2 ANALYZING THE INDUCTIVE BIAS OF DIFFUSION MODELS

Diffusion models are designed to map a Gaussian noise distribution to a dataset distribution. To achieve this, diffusion models take a noisy image $\boldsymbol{x}_t$, obtained by adding Gaussian noise $\boldsymbol{\epsilon}$ to a training sample $\boldsymbol{x}_0$ following a noise schedule depending on step $t$, and estimate noise $\boldsymbol{\epsilon}$. The loss function of diffusion model training is as follows:

$$\mathcal{L} = \mathbb{E}_{\boldsymbol{x}_0, \boldsymbol{\epsilon}, t} \left[ \| \boldsymbol{\epsilon} - \boldsymbol{\epsilon}_\theta(\boldsymbol{x}_t, t) \|_2^2 \right]. \tag{1}$$

In Eq. (1), $\boldsymbol{\epsilon}_\theta(\cdot)$ represents the backbone network with trainable parameters $\theta$, which plays a crucial role in diffusion model generalization and hence is our primary focus. In this section, we first compare the generalization ability of a DiT (Peebles & Xie, 2023) and a UNet (Nichol & Dhariwal, 2021), two of the most popular diffusion model backbones. Subsequently, we investigate the inductive biases that drive their generalization.

### 2.1 COMPARING DIT AND UNET GENERALIZATION

We compare the generalization of pixel-space DiT and UNet[1] using as a metric the PSNR gap proposed by Kadkhodaie et al. (2024). The PSNR gap is the zero-truncated difference between the

---

[1]https://github.com/openai/improved-diffusion

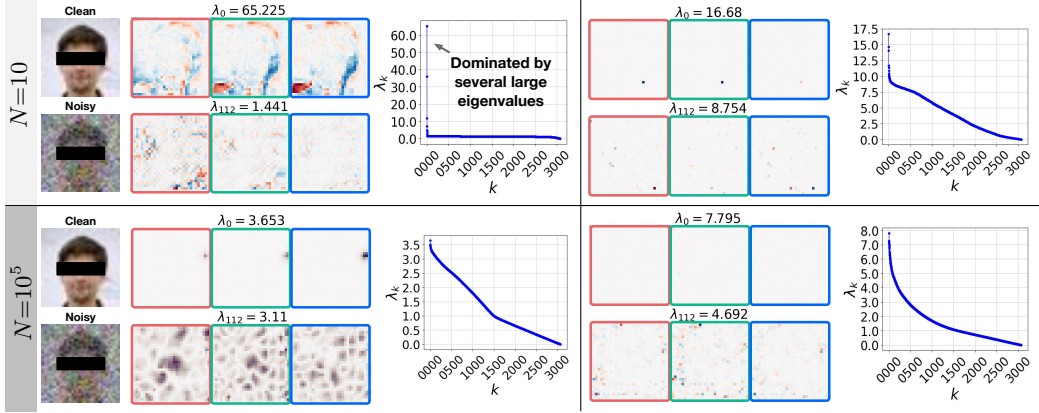

(a) UNet, FLOPs: 303.17G; Params: 109.55M  (b) DiT, FLOPs: 300.49G; Params: 14.27M

Figure 3: Jacobian eigenvector comparison between UNet (Nichol & Dhariwal, 2021) and DiT (Peebles & Xie, 2023) with equivalent FLOPs. (a) The eigenvectors of a UNet tend to memorize the training images when $N=10$ and drive the generalization through harmonic bases (Kadkhodaie et al., 2024) when $N=10^5$. In contrast, (b) the DiT's eigenvectors exhibit neither the memorization effect at $N=10$ nor harmonic bases at $N=10^5$.

training set PSNR and the testing set PSNR at a diffusion step $t$:

$$\text{PSNR Gap}(t) = \max\left(\text{PSNR}_{\text{train}}(t) - \text{PSNR}_{\text{test}}(t), 0\right), \quad (2)$$

where $\text{PSNR}_{\text{train}}(t)$ and $\text{PSNR}_{\text{test}}(t)$ are obtained following Kadkhodaie et al. (2024). To elaborate, given $K$ images from either training or testing set, we first feed noisy images at step $t$ to diffusion models and obtain the estimated noise $\hat{\boldsymbol{\epsilon}}$. Next, we get the one-step denoising result $\hat{\boldsymbol{x}}_0$ via

$$\hat{\boldsymbol{x}}_0 = \boldsymbol{x}_t - \sigma_t \hat{\boldsymbol{\epsilon}}, \quad (3)$$

where $\sigma_t$ is defined by the diffusion model noise scheduler. Finally, we derive the training and testing PSNRs at diffusion step $t$ as follows:

$$\text{PSNR}_{\text{train/test}}(t) = 10 \cdot \left(\log(M^2) - \log\left(\frac{1}{K}\sum_{k=1}^{K}\text{MSE}\left(\hat{\boldsymbol{x}}_0^k, \boldsymbol{x}_0^k\right)\right)\right). \quad (4)$$

Here, $\hat{x}_0^k$ denotes the estimated $x_0$ for image $k$ at diffusion step $t$ obtained by using Eq. (3), $M$ denotes the intensity range of $\boldsymbol{x}_0$, which is set to 2 since $\boldsymbol{x}_0$ is normalized to $[-1, 1]$. $K$ is set to 300 following the PSNR gap computation of Kadkhodaie et al. (2024).

Turning to diffusion model backbones, prior work (Peebles & Xie, 2023) has shown that a DiT achieves better image generation quality than a UNet with equivalent FLOPs. This advantage of DiT prompts our curiosity to study whether DiT can also demonstrate superiority in generalization, using the PSNR gap as a metric. Fig. 2 compares the PNSR and PSNR gap of a UNet and a DiT. Interestingly, when the number of training images is sufficient for the model size, e.g., $N=10^5$, the training and testing PSNR curves of both DiT and UNet are nearly identical, and their PSNR gaps remain small. This indicates that DiT and UNet have no substantial performance difference in distribution mapping given sufficient training data. Nevertheless, as shown in Fig. 2 (b), when trained with less data, e.g., $N=10^3$ and $N=10^4$, a DiT has a remarkably smaller PSNR gap than a UNet, suggesting that a DiT has a better generalization ability than a UNet. This discrepancy of the PSNR gap motivates us to explore the underlying inductive biases that contribute to the generalization difference between a DiT and a UNet.

## 2.2 DiT Does Not Have Geometry-Adaptive Harmonic Bases

Kadkhodaie et al. (2024) reveal that the generalization of a simplified one-channel UNet is driven by the emergence of geometry-adaptive harmonic bases. These harmonic bases are obtained from the eigenvectors of a UNet's Jacobian matrix. This raises an important question: can the potential difference in harmonic bases between a DiT and a UNet account for their generalization differences? To address this, we follow the approach of Kadkhodaie et al. (2024) and perform an eigendecomposition of the Jacobian matrices for a three-channel classic UNet (Nichol & Dhariwal, 2021) and

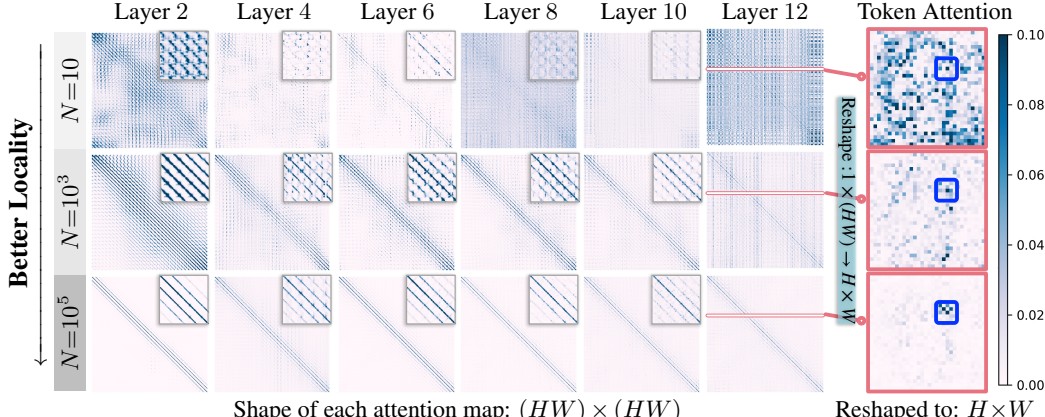

Figure 4: Attention maps of DiTs trained with $10$, $10^3$, and $10^5$ images. All attention maps are linearly normalized to the range $[0, 1]$, with a colormap applied to the interval $[0, 0.1]$ for enhanced visualization. The top-right insets provide a zoomed-in view of the center patch of each attention map. As the number of training images increases, DiT's generalization improves, and attention maps across all layers exhibit stronger locality. The pink boxes highlight the attention corresponding to a specific output token, obtained by reshaping a single row from the layer-12 attention map (original shape: $1 \times (HW)$) into a matrix of shape $H \times W$. As $N$ increases from $10$ to $10^5$, the token attentions progressively concentrate around the region near the output token (highlighted with blue boxes).

a DiT. Specifically, we first feed a noisy image $x$ ($x_t$, $t$ is omitted for simplicity) into a DiT and a UNet and obtain their Jacobian matrices, where each entry of the Jacobian

$$
\nabla \boldsymbol{\epsilon}_\theta = \begin{bmatrix} \frac{\partial \hat{\epsilon}_1}{\partial x_1} & \frac{\partial \hat{\epsilon}_1}{\partial x_2} & \cdots & \frac{\partial \hat{\epsilon}_1}{\partial x_{HW}} \\ \frac{\partial \hat{\epsilon}_2}{\partial x_1} & \frac{\partial \hat{\epsilon}_2}{\partial x_2} & \cdots & \frac{\partial \hat{\epsilon}_2}{\partial x_{HW}} \\ \vdots & \vdots & \ddots & \vdots \\ \frac{\partial \hat{\epsilon}_{HW}}{\partial x_1} & \frac{\partial \hat{\epsilon}_{HW}}{\partial x_2} & \cdots & \frac{\partial \hat{\epsilon}_{HW}}{\partial x_{HW}} \end{bmatrix}, \quad \hat{\boldsymbol{\epsilon}} = \boldsymbol{\epsilon}_\theta(\boldsymbol{x}, t), \quad \boldsymbol{x}, \hat{\boldsymbol{\epsilon}} \in \mathbb{R}^{(HW) \times d}, \tag{5}
$$

represents the partial derivative of an output pixel w.r.t. all input pixels. Next, we perform an eigen-decomposition of the Jacobian matrix and obtain the eigenvectors.

Fig. 3 presents the eigenvalues and eigenvectors of a UNet and a DiT trained with $10$ and $10^5$ images, respectively. For a UNet which is trained with a small number of images, *e.g.*, $N{=}10$, the Jacobian eigenvectors corresponding to several large eigenvalues tend to memorize the geometry of the input image. Notably, the leading eigenvalues are significantly larger than the rest, indicating that the UNet trained with 10 images is governed by memorization of the training images (Carlini et al., 2023; Somepalli et al., 2023). In contrast, when the training set size is increased to $N{=}10^5$, the UNet's eigenvectors show the geometry-adaptive harmonic bases similar to the ones reported by Kadkhodaie et al. (2024): oscillating patterns which increase in frequency as eigenvalues $\lambda_k$ decrease. This clear transition from memorizing to generalizing, observed as $N$ increases, indicates that harmonic bases play a key role in driving the generalization of a UNet.

In contrast, harmonic bases do not appear to be the driving factor behind a DiT's generalization. As shown in Fig. 3 (b), the eigenvectors of the DiT do not exhibit the harmonic bases similar to the ones observed for the UNet. Instead, the DiT displays random sparse patterns regardless of the training dataset size. Additionally, the difference in the distribution of DiT's eigenvalues between $N{=}10$ and $N{=}10^5$ is much less pronounced compared to that of the UNet. Notably, unlike the UNet, the Jacobian eigenvectors of the DiT does not transition from memorization to generalization as the training dataset size increases, indicating that the driving factor of a DiT's generalization is fundamentally different from a UNet. This difference calls for a further study about what other inductive biases drive the generalization ability of a DiT?

## 2.3 How Does a DiT Generalize?

The generalization of a DiT may originate from the self-attention (Vaswani, 2017) dynamics because of its pivotal role in a DiT. Could the attention maps of a DiT provide insights into its inductive

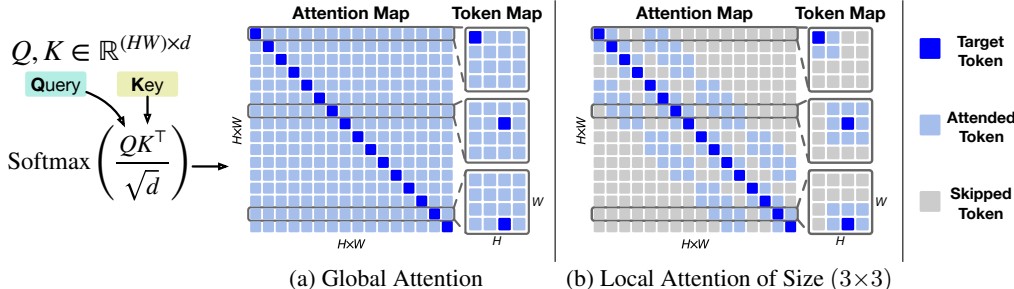

(a) Global Attention      (b) Local Attention of Size $(3{\times}3)$

Figure 5: Global and local attention maps: (a) global attention captures the relationship between the target token and any input token, whereas (b) local attention focuses only on tokens within a nearby window around the target.

biases? In light of this, we empirically compare the attention maps of DiTs with varying levels of generalization: three DiT models trained with $10$, $10^3$, and $10^5$ images, where a DiT trained with more images demonstrates stronger generalization. Specifically, we extract and visualize the attention maps from the self-attention layers of these DiT models as follows,

$$\text{Attention Map} = \text{Softmax}\left(\frac{QK^\top}{\sqrt{d}}\right), \ \ \{Q,K\} \in \mathbb{R}^{(HW)\times d}, \tag{6}$$

where $Q$ and $K$ represent the query and key matrices. $H$ and $W$ are the height and width of the input tensor, while $d$ denotes the dimension of a self-attention layer. For better readability of the attention maps, we linearly normalize each attention map to the range of $[0,1]$ and apply a colormap to the interval $[0, 0.1]$, *i.e.*, values exceeding the upper bound are clipped at $0.1$.

Fig. 4 shows the attention maps of DiTs with varying levels of generalization on a randomly selected image. Empirically, we observe that the attention maps of a DiT's self-attention layers remain highly consistent across different images. Further details are provided in Appendix B. As the number of training images increases from $N{=}10$ to $N{=}10^5$, the attention maps of a DiT become increasingly concentrated along several diagonal lines. A closer inspection of the token attentions of a specific target token, *i.e.*, a row in the attention map, shows that these diagonal patterns correspond to tokens near the target token, indicating that the generalization ability of a DiT is linked to the locality of its attention maps. Delving deeper, can one modify the generalization of a DiT with this inductive bias? We explore this next.

## 3   INJECTING INDUCTIVE BIAS BY RESTRICTING ATTENTION WINDOWS

To verify that the locality of attention maps enables the generalization of a DiT, we hypothesize that it's possible to adjust the inductive bias of a DiT by restricting attention windows. To test this hypothesis, we set up baselines by adopting the diffusion model and DiT implementations from the official repository[2] of Peebles & Xie (2023). Specifically, we remove the auto-encoder and set the patchify size to $1{\times}1$, transforming it into a pixel-space DiT. This modification rules out irrelevant components and ensures more straightforward comparisons in downstream experiments. For model training, we use images of resolution $32{\times}32$, which is equivalent in dimensionality to $512{\times}512$ for a latent-space DiT with a patchify size of $2{\times}2$.

In the remainder of this section, we show that based on the PSNR gap, injecting local attention can effectively modify a DiT generalization, often accompanied by an FID change when insufficient training data is used. Furthermore, we discover that placing the attention window restrictions at different locations in a DiT and adjusting the effective attention window sizes allow for additional control over its generalization behavior. Details *w.r.t.* experimental settings and more results are deferred to the Appendix A and D.

---

[2] https://github.com/facebookresearch/DiT

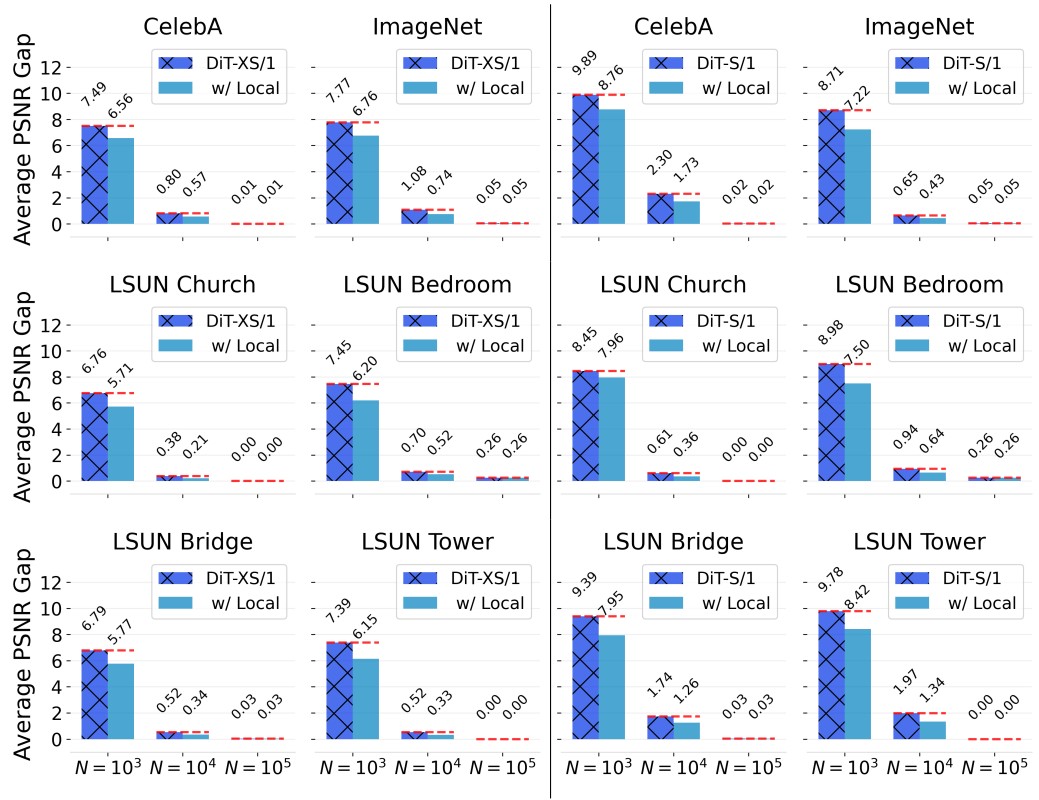

(a) DiT-XS/1, hidden_size: 252, num_heads: 4    (b) DiT-S/1, hidden_size: 384, num_heads: 6

Figure 6: PSNR gap↓ comparison between a standard DiT and a DiT equipped with local attention for two architectures: (a) DiT-XS/1 and (b) DiT-S/1. Incorporating local attention reduces the PSNR gap consistently across $N=10^3$, $N=10^4$, and $N=10^5$. This advantage is robust across six different datasets and both DiT backbones. In this setup, local attention with window sizes $(3, 5, 7, 9, 11, 13)$ is applied to the first six layers of the DiT. Textured bars highlight the default DiT baselines.

## 3.1 ATTENTION WINDOW RESTRICTION

Local attention, initially proposed to enhance computational efficiency (Liu et al., 2021; Yang et al., 2022; Hatamizadeh et al., 2023; Hassani et al., 2023), is a straightforward yet effective way to modify a DiT's generalization. Different from global attention which enables a target token to connect with all input tokens (Fig. 5 (a)), local attention only permits a target token to attend within a small nearby window. The resulting attention map structure is depicted in Fig. 5 (b). Notably, a local attention constrains the attention map to a sparse activation pattern only along the diagonal direction, thereby enforcing locality of the attention map. The resulting attention map patterns produced by a local attention align well with the inductive bias that a DiT exhibits when observing a strong generalization ability, as illustrated in Fig. 4 (row $N=10^5$).

Using local attentions in a DiT can consistently improve its generalization (measured by PSNR gap) across different datasets and model sizes. Specifically, we consider a DiT model with 12 DiT blocks, and replace the first 6 global attention layers with local attentions, whose window sizes range from $3\times3$ to $13\times13$ with a stride of 2. We train both the vanilla DiT and a DiT equipped with local attentions with $N=10^3, 10^4$ and $10^5$ images for the same 400k training steps. Then we calculate the PSNR gap between the training and testing images for models trained with different amounts of images. In Fig. 6, we show the PSNR gap comparison between a DiT with and without local attentions on CelebA, ImageNet, and LSUN (Church, Bedroom, Bridge, Tower) datasets, using baseline DiT models of two sizes (DiT-XS/1 and DiT-S/1). Notably, using local attentions reduces a DiT's PSNR gap with different amounts of training images. Importantly, the advantage of local attention is robust across different training datasets and backbone sizes.

Table 1: FID↓ comparison between a standard DiT and a DiT equipped with local attention. $^\dagger$ indicates training with different random seeds, train-test splits, and doubled batch sizes. For the DiT-XS/1 and DiT-S/1 architectures, local attention reduces FID when the DiT's generalization is not saturated ($N=10^4$). At $N=10^5$, local attention achieves comparable or marginally higher FID compared to the standard DiT. These findings are consistent across various datasets, random seeds, train-test splits, and batch sizes. In this setting, local attention with window sizes of $(3, 5, 7, 9, 11, 13)$ is applied to the first six layers of the DiT, where both the placement and window size play a crucial role in determining a DiT's FID result. Further details are provided in Sec. 3.2 and Sec. 3.3.

| Model | CelebA | | ImageNet | | LSUN Church | | LSUN Bedroom | | LSUN Bridge | | LSUN Tower | |
|---|---|---|---|---|---|---|---|---|---|---|---|---|
| | $N=10^4$ | $N=10^5$ | $N=10^4$ | $N=10^5$ | $N=10^4$ | $N=10^5$ | $N=10^4$ | $N=10^5$ | $N=10^4$ | $N=10^5$ | $N=10^4$ | $N=10^5$ |
| DiT-XS/1 | 9.6932 | 2.6303 | 52.5650 | 17.3114 | 12.8842 | 5.2927 | 14.8354 | 5.4066 | 23.1771 | 8.0791 | 12.5532 | 4.6619 |
| w/ Local | 8.4580 | 2.5469 | 43.8687 | 18.0671 | 10.4794 | 5.2672 | 11.9566 | 5.3542 | 18.1470 | 8.3546 | 10.5644 | 4.8041 |
| | $-12.74\%$ | $-3.17\%$ | $-16.54\%$ | $+4.37\%$ | $-18.66\%$ | $-0.97\%$ | $-19.40\%$ | $-0.97\%$ | $-21.70\%$ | $+3.41\%$ | $-15.84\%$ | $+3.05\%$ |
| DiT-XS/1$^\dagger$ | 10.5432 | 2.5215 | 36.8461 | 20.1907 | 13.4921 | 3.9033 | 15.6740 | 4.8256 | 22.0032 | 7.7771 | 13.8952 | 4.1576 |
| w/ Local$^\dagger$ | 8.4258 | 2.4988 | 31.4555 | 20.3175 | 10.2708 | 4.5322 | 11.2033 | 5.0868 | 17.8903 | 7.7477 | 10.1938 | 4.6146 |
| | $-20.08\%$ | $-0.90\%$ | $-14.63\%$ | $+0.63\%$ | $-23.88\%$ | $+16.11\%$ | $-28.53\%$ | $+5.41\%$ | $-18.69\%$ | $-0.38\%$ | $-26.64\%$ | $+10.99\%$ |
| DiT-S/1 | 23.2496 | 2.3278 | 36.6378 | 20.6101 | 14.8826 | 3.9390 | 16.1094 | 4.6086 | 51.5729 | 5.7950 | 28.9727 | 3.1897 |
| w/ Local | 20.7768 | 2.3321 | 33.1807 | 20.7972 | 11.7540 | 4.4097 | 11.6833 | 5.0519 | 37.6523 | 5.5825 | 21.8068 | 3.5586 |
| | $-10.64\%$ | $+0.18\%$ | $-9.44\%$ | $+0.91\%$ | $-21.02\%$ | $+11.95\%$ | $-27.48\%$ | $+9.62\%$ | $-26.99\%$ | $-3.67\%$ | $-24.73\%$ | $+11.57\%$ |
| DiT-S/1$^\dagger$ | 14.1763 | 2.5061 | 37.3477 | 20.4165 | 15.4509 | 4.2317 | 15.5820 | 4.8336 | 24.4374 | 7.3170 | 14.8695 | 4.4495 |
| w/ Local$^\dagger$ | 11.1046 | 2.6598 | 33.1323 | 20.6006 | 11.4956 | 4.5546 | 11.3673 | 5.0552 | 20.3403 | 7.5565 | 12.3236 | 4.4927 |
| | $-21.67\%$ | $+6.13\%$ | $-11.29\%$ | $+0.90\%$ | $-25.60\%$ | $+7.63\%$ | $-27.05\%$ | $+4.58\%$ | $-16.77\%$ | $+3.27\%$ | $-17.12\%$ | $+0.97\%$ |

For a discriminative model, *e.g.*, a classifier, better generalization typically leads to better model performance when the training dataset is insufficient. Is this also the case for generative models like a DiT? To investigate, we compare the FID between the default DiT and a DiT using local attentions. For each dataset, we compare FID values of models trained with $10^4$ and $10^5$ images: the former represents the case of insufficient training images while the later case refers to use of sufficient training data. Tab. 1 shows the FID comparison among the same six datasets and two DiT backbones used when comparing PSNR gaps. Improving the generalization via local attentions can indeed improve the FID when $N=10^4$, which is in line with observations from discriminative models. When $N=10^5$, using the presented approach of adding local attentions either results in comparable FID values or experiences a slight compromise. Interestingly, we find that modifying the placement and effective attention window size permits fine-grained control of a DiT's generalization and generation quality. More discussions are in Sec. 3.2 and Sec. 3.3 below.

In light of Occam's razor, reducing the model parameter count has been shown to be yet another possible strategy to inject an inductive bias. This differs from the attention window restrictions considered above, as local attentions reduce the FLOPs of a DiT without changing the model parameter count. In contrast, to inject an inductive bias by reducing the parameter count of a DiT, we explore sharing of the parameters of a DiT's attention blocks as well as modifying a DiT's attention layers to learn the coefficients of pre-computed offline PCA components. Neither of these methods shows as compelling improvements of the generalization (measured via the PSNR gap) as using local attentions. We provide more details regarding the considered techniques in the Appendix C.

## 3.2 PLACEMENT OF ATTENTION WINDOW RESTRICTION

Given the same set of local attentions, placing them at different layers of a DiT leads to different results. For local attention, we study three placement schemes: 1) placing local attentions on the early layers of a DiT, 2) interleaving local attentions with global attentions, and 3) placing local attentions on the tail layers of a DiT. In Fig. 7, we compare the PSNR gap for the three aforementioned placement schemes on the CelebA and ImageNet datasets, using two distinct local attention configurations. Specifically, *Local* refers to a setting with 6 attention layers, where the window sizes vary from $3\times3$ to $13\times13$ with a stride of 2, which is consistent with the local attention configuration used in Fig. 6 and Tab. 1 above. Meanwhile, *Local*$^*$ represents a different configuration consisting of 9 local attention layers, arranged as $(3^{*3}, 5^{*3}, 7^{*3})$, where $i^{*j}$ indicates repeating a local attention layer with a $(i\times i)$ window $j$ times.

The results in Fig. 7 indicate that applying local attention in the early layers of a DiT consistently leads to a smaller PSNR gap across different training data sizes. Additionally, the FID results in

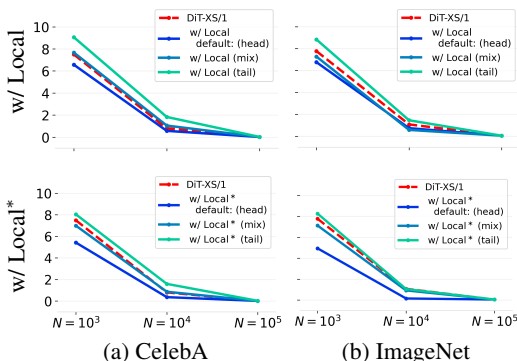

(a) CelebA      (b) ImageNet

Figure 7: PSNR gap↓ comparison for different local attention placement patterns. We find that placing local attention in the early layers (head) results in a smaller PSNR gap compared to mixing local and global attention (mix) or applying local attention in the later layers (tail). The latter two configurations may even perform worse than the vanilla DiT.

Table 2: FID↓ comparison for different local attention placement patterns. Local* represents using nine local attention layers with window sizes $\left(3^{*3}, 5^{*3}, 7^{*3}\right)$ in a DiT. Placing local attention in the early layers achieves lower FIDs when $N=10^4$, indicating successful generalization modification. In contrast, mix and tail placements fail to consistently modify the generalization of a DiT. The lowest FIDs are highlighted in **bold**.

| Model | CelebA | | ImageNet | |
|---|---|---|---|---|
| | $N=10^4$ | $N=10^5$ | $N=10^4$ | $N=10^5$ |
| DiT-XS/1 | 9.6932 | 2.6303 | 52.5650 | 17.3114 |
| w/ Local (head) | **8.4580** | 2.5469 | 43.8687 | 18.0671 |
| w/ Local (mix) | 11.8858 | 2.5015 | **37.6397** | 18.4266 |
| w/ Local (tail) | 18.0717 | **2.4288** | 59.8510 | **17.5818** |
| w/ Local* (head) | **7.2307** | 3.0991 | **29.2520** | 23.7896 |
| w/ Local* (mix) | 10.9537 | **2.7068** | 51.8233 | **18.7975** |
| w/ Local* (tail) | 17.0445 | 3.0400 | 49.6403 | 22.1723 |

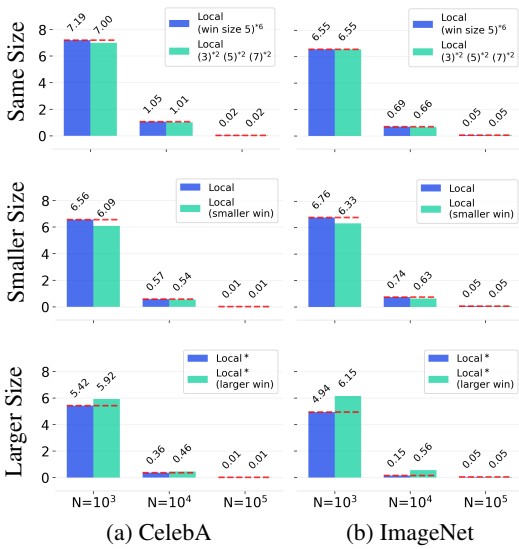

(a) CelebA      (b) ImageNet

Figure 8: PSNR gap↓ changes when the effective attention window size is kept constant, decreased, or increased. Reducing the window size results in a smaller PSNR gap, indicating improved generalization.

Table 3: FID↓ changes when the effective attention window size is kept constant, decreased, or increased. Modifying the attention window distribution while keeping the overall window size unchanged results in minimal FID changes when $N=10^4$. Decreasing the window size improves generalization, leading to lower FID at $N=10^4$, whereas increasing the window size has the opposite effect.

| Model | CelebA | | ImageNet | |
|---|---|---|---|---|
| | $N=10^4$ | $N=10^5$ | $N=10^4$ | $N=10^5$ |
| Local Attn ($5^{*6}$) | 12.9798 | 2.3348 | 40.7373 | 17.8686 |
| $(3^{*2}, 5^{*2}, 7^{*2})$ | 12.6680 | 2.3455 | 40.7499 | 17.7538 |
| | $-2.40\%$ | $+0.46\%$ | $+0.03\%$ | $0.64\%$ |
| Local | 8.4580 | 2.5469 | 43.8687 | 18.0671 |
| (smaller win size) | 8.0543 | 2.7174 | 39.5779 | 18.9400 |
| | $-4.77\%$ | $+6.69\%$ | $-9.78\%$ | $+4.83\%$ |
| Local* | 7.2307 | 3.0991 | 29.2520 | 23.7896 |
| (larger win size) | 7.8800 | 2.8577 | 37.8708 | 19.3568 |
| | $+8.98\%$ | $7.79\%$ | $+29.46\%$ | $18.63\%$ |

Tab. 2 demonstrate that the first placement scheme generally improves FID when the training data is limited ($N=10^4$). In contrast, interleaving local and global attention, or applying local attention to the tail layers, enhances the model's data-fitting ability but often compromises generalization. These two placement schemes tend to improve FID when $N=10^5$, though this improvement comes at the cost of reduced FID when $N=10^4$, further supporting the generalization results as measured by the PSNR gap.

### 3.3 Effective Attention Window Size

Adjusting the effective attention window size provides an additional mechanism to control the generalization of a DiT. Specifically, our analysis reveals that smaller attention windows lead to stronger

generalization, while larger windows enhance data fitting, typically at the cost of generalization. Furthermore, maintaining the total attention window size but altering the distribution across local attentions generally preserves the overall behavior of a DiT. These observations are based on an empirical study using the CelebA and ImageNet datasets, involving three paired comparisons of local attention configurations. The PSNR gap and FID results are shown in Fig. 8 and Tab. 3, respectively.

Specifically, in the first comparison, we apply two configurations of local attentions with window sizes $(5, 5, 5, 5, 5, 5)$ and $(3, 3, 5, 5, 7, 7)$ to the first six layers of a DiT. We observe that altering the attention window size distribution, while keeping the total window size fixed, has a limited impact on a DiT's generalization, as indicated by the similar PSNR gaps across $N=10^3$, $10^4$, and $10^5$. This similarity in generalization is further corroborated by their comparable FID values. In the second and third comparisons, using the DiT-XS/1 configurations with *Local* and *Local** attention settings, we find that reducing the attention window size enhances generalization, while increasing the window size diminishes it. This is evidenced by a decrease in the PSNR gap for smaller window sizes and an increase for larger ones. Furthermore, the improved generalization is associated with better FID values under comparably insufficient training data, and vice versa.

## 4 RELATED WORK

**Inductive Biases of Generative Models.** Current diffusion models (Sohl-Dickstein et al., 2015; Song et al., 2020; Ho et al., 2020; Kadkhodaie & Simoncelli, 2020; Nichol & Dhariwal, 2021; Song et al., 2020; An et al., 2024) exhibit strong generalization abilities (Zhang et al., 2021; Keskar et al., 2016; Griffiths et al., 2024; Wilson & Izmailov, 2020), relying on inductive biases (Mitchell, 1980; Goyal & Bengio, 2022). Prior to the emergence of diffusion models, Zhao et al. (2018) show that generative models like GANs (Goodfellow et al., 2020) and VAEs (Kingma, 2013) can generalize to novel attributes not presented in the training data. This generalization ability of generative models is possibly due to the inductive biases (Zhang et al., 2021; Keskar et al., 2016) introduced by model design and training. Following this line of research, Kadkhodaie et al. (2024) show that the generalization of diffusion models arises due to geometry-adaptive harmonic bases (Mallat et al., 2020). However, their work only studies the generalization of a simplified one-channel UNet. It remains unclear whether their study can be generalized to commonly used three-channel UNets (Nichol & Dhariwal, 2021) and more compelling DiTs (Peebles & Xie, 2023). This work fills this gap and reveals that a classic UNet still exhibits the harmonic bases but a DiT does not. Further studies show that a DiT's generalization is associated with a different inductive bias: locality of attention maps.

**Attention Window Restrictions.** Prior studies have shown that restricting attention windows through mechanisms such as local attention (Beltagy et al., 2020; Liu et al., 2021; Hassani et al., 2023), strided attention (Wang et al., 2021; Xia et al., 2022), and sliding attention (Pan et al., 2023), among others, can significantly improve the efficiency of attention computation (Yang et al., 2022; Hatamizadeh et al., 2023; Hassani et al., 2023; Apple, 2024). These techniques limit the attention scope, reducing computational complexity while retaining the model's ability to capture important contextual information. However, our work explores a different direction by investigating how attention window restrictions, especially through local attention, affect the generalization properties of DiTs. We show that beyond efficiency gains, local attention can be used to modulate the model's generalization by enforcing the inductive bias of locality within attention maps.

## 5 CONCLUSION

This paper investigates the inductive biases that facilitate the generalization ability of DiTs. For insufficient training data, we observe that DiTs achieve superior generalization, as measured by the PSNR gap, compared to UNets with equivalent FLOPs. However, unlike simplified and standard UNet-based diffusion models, DiTs do *not* exhibit geometry-adaptive harmonic bases. Motivated by this discrepancy, we explore alternative inductive biases and identify that a DiT's generalization is instead influenced by the locality of its attention maps. Consequently, we effectively modulate the generalization behavior of DiTs by incorporating local attention layers. Specifically, we demonstrate that varying the placement of local attention layers and adjusting the effective attention window size enables fine-grained control of a DiT's generalization and data-fitting capabilities. Enhancing a DiT's generalization often leads to improved FID scores when trained with insufficient data.

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
