# Supplementary Material: On Inductive Biases That Enable Generalization of Diffusion Transformers

## Contents

## A Experimental Settings

### A.1 Model Definition

We verify the effectiveness of local attention in modifying the generalization of a DiT using 2 DiT backbones and 10 local attention variations. Tab. 1 provides more details about the DiT backbones and local attention configurations.

To elaborate, we adopt two DiT backbones, DiT-XS/1 and DiT-S/1, to verify the effectiveness of attention window restrictions in modifying the generalization of a DiT. Both models have 12 DiT

Table 1: DiT architectures and local attention settings. In the column titled 'DiT Blocks', $G$ denotes global attention while a number $k$ represents a local attention of window size $k \times k$.

| Model | DiT Blocks | Hidden Size | Num Heads | Depth | Patch Size |
|---|---|---|---|---|---|
| DiT-XS/1 | $(G, G, G, G, G, G, G, G, G, G, G, G)$ | 252 | 4 | 12 | $1 \times 1$ |
| w/ Local | $(3, 5, 7, 9, 11, 13, G, G, G, G, G, G)$ | 252 | 4 | 12 | $1 \times 1$ |
| w/ Local (mix) | $(3, G, 5, G, 7, G, 9, G, 11, G, 13, G)$ | 252 | 4 | 12 | $1 \times 1$ |
| w/ Local (tail) | $(G, G, G, G, G, G, 3, 5, 7, 9, 11, 13)$ | 252 | 4 | 12 | $1 \times 1$ |
| w/ Local (smaller win) | $(3, 5, 7, 9, 11, 13, 15, 17, 19, G, G, G)$ | 252 | 4 | 12 | $1 \times 1$ |
| w/ Local* | $(3, 3, 3, 5, 5, 5, 7, 7, 7, G, G, G)$ | 252 | 4 | 12 | $1 \times 1$ |
| w/ Local* (mix) | $(3, 3, 3, G, 5, 5, 5, G, 7, 7, 7, G)$ | 252 | 4 | 12 | $1 \times 1$ |
| w/ Local* (tail) | $(G, G, G, 3, 3, 3, 5, 5, 5, 7, 7, 7)$ | 252 | 4 | 12 | $1 \times 1$ |
| w/ Local* (larger win) | $(5, 5, 5, 7, 7, 7, 9, 9, 9, G, G, G)$ | 252 | 4 | 12 | $1 \times 1$ |
| w/ Local Attn $(5^{*6})$ | $(5, 5, 5, 5, 5, 5, G, G, G, G, G, G)$ | 252 | 4 | 12 | $1 \times 1$ |
| w/ Local Attn $(3^{*2}, 5^{*2}, 7^{*2})$ | $(3, 3, 5, 5, 7, 7, G, G, G, G, G, G)$ | 252 | 4 | 12 | $1 \times 1$ |
| DiT-S/1 | $(G, G, G, G, G, G, G, G, G, G, G, G)$ | 384 | 6 | 12 | $1 \times 1$ |
| w/ Local | $(3, 5, 7, 9, 11, 13, G, G, G, G, G, G)$ | 384 | 6 | 12 | $1 \times 1$ |
| DiT-XXS/1 | $(G, G, G, G, G, G, G, G, G, G, G, G)$ | 240 | 4 | 12 | $1 \times 1$ |

**Blocks.** We remove the auto-encoder and use a patch size of $1 \times 1$. Note, DiT-XS/1 has a hidden size of 252 and uses 4 attention heads. In contrast, DiT-S/1 has a hidden size of 384 and uses 6 attention heads. Regarding the local attention variations, the default setting *Local* combines 6 local attentions of window size $(3, 5, 7, 9, 11, 13)$ and 6 global attentions. Meanwhile, *Local** is a variant using 9 local attentions of window size $(3, 3, 3, 5, 5, 5, 7, 7, 7)$. For both *Local* and *Local** settings, we place local attentions at the heading layers of a DiT. We also study interleaving the local and global attentions as well as placing local attentions at tailing layers of a DiT, leading to (mix) and (tail) variants. Additionally, to study the effects of modifying the attention window size, we decrease the attention window size of the *Local* model and increase the attention window size of the *Local** model, resulting in (smaller win) and (larger win) models in Tab. 1.

## A.2 TRAINING AND SAMPLING SETTING

The implementation of the UNet[*] and the DiT[†] are based on the official repositories of Nichol & Dhariwal (2021) and Peebles & Xie (2023). Specifically, for the UNet, we use the architecture which has a 4-stage encoder with channel multipliers of $(1, 2, 3, 4)$. For each stage, we include 3 ResBlocks. At the end of each stage, the resolution of the input tensor are down-sampled by a factor of 2. In the last stage, we use one layer of self-attention. The decoder mirrors the encoder layers and places them in the reverse order, replacing down-sampling layers with up-sampling ones. Between the encoder and decoder, there are 2 ResBlocks and 1 self-attention layer by default. The default skip connections are used between the encoder and decoder at the same resolution. Consequently, the UNet has 303.17G FLOPs and 109.55M parameters. The FLOPs of this UNet are nearly identical to the DiT-XS/1 model in Tab. 1.

All DiT and UNet models are trained with the same hyper-parameter settings. Concretely, we train each model in the pixel-space using a resolution of $32 \times 32$. All networks are using the same diffusion algorithm: diffusion steps of 1000 in training and 250 in sampling, and predicting the added noise and sigma simultaneously. To train a network, we use the random seed 43, learning rate $1e^{-4}$ and an overall batch size of 64. All networks are trained with 8 or 4 A100/H100 GPUs, using the EMA checkpoint at train step $400k$ with EMA decay 0.9999. For each dataset, we first randomly shuffle the whole dataset. Then we choose the last $N = 10, 10^3, 10^4$ and $10^5$ images as the training set. The train-test split of a dataset is kept consistent for different networks. When computing FID values for the model trained with $N = 10^4$ and $10^5$ images, we randomly select $M = \min\{N, 50k\}$ of the training images as the reference set. In Tab. 1 of the main manuscript, we present the results of

---

[*] https://github.com/openai/improved-diffusion

[†] https://github.com/facebookresearch/DiT

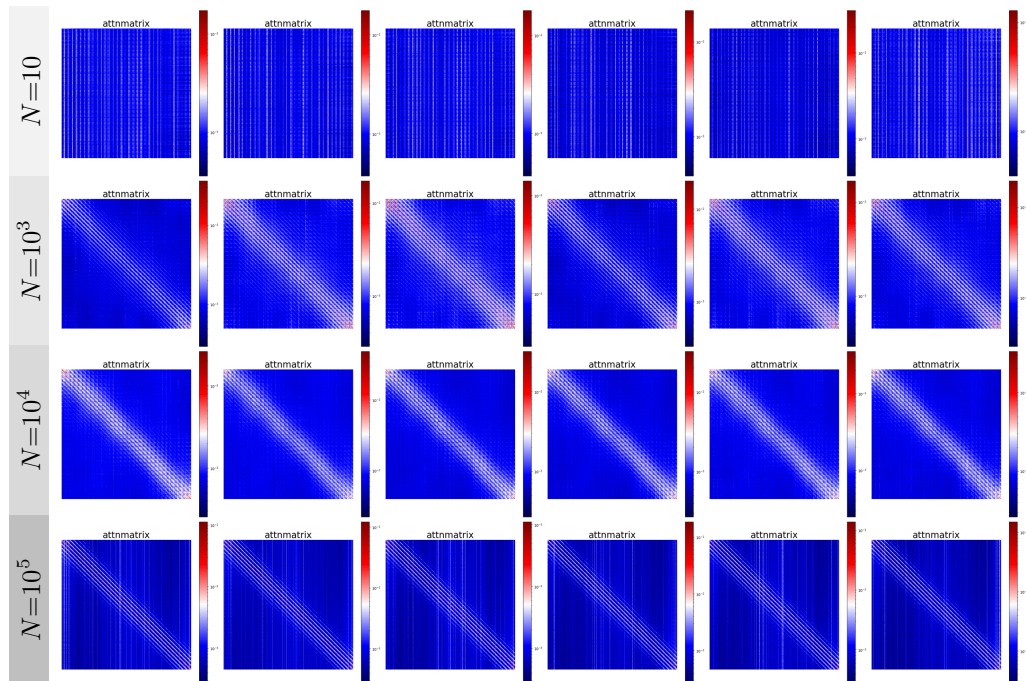

Figure 1: Attention maps of the $1^{st}$ DiT Block produced by feeding 6 random images to DiT-XS/1 models.

DiT-XS/1$^\dagger$ and DiT-S/1$^\dagger$ models, with and without local attentions. For the $^\dagger$ models, we use a different dataset shuffling, change the random seed to 143, and double the training batch size to 128. Notably, using local attention on both normal and $^\dagger$ models can successfully modify the generalization of a DiT, confirming the effectiveness of the locality as the inductive bias of a DiT.

### A.3 PSNR COMPUTATION

We compute the PSNR based on a training or testing subset of 300 images following Kadkhodaie et al. (2024). For each image, we re-space the diffusion steps from 1000 to 50, and compute the train and test PSNR on each step. Specifically, we first perform the noising step of the diffusion model to get the noisy image at a diffusion step $t$. Next, we feed the noisy image into the diffusion model backbone and get the estimation of the added noise, which is then used to recover the clean image from the training or testing subset, i.e., performing a one-step denoising. The final PSNR at step $t$ is obtained using the estimated clean image and the ground truth. Consequently, the PSNR value can estimate a diffusion model's accuracy at each diffusion step. Therefore, the PSNR gap between the training and testing subsets can measure a diffusion model's generalization: when a diffusion model has good generalization, its prediction accuracy should be comparably between the training and testing set, resulting in a small PSNR gap.

### B ATTENTION MAP CONSISTENCY

To verify the robustness of the discovered inductive bias of a DiT, i.e., the locality of attention maps, we obtain the attention maps corresponding to distinct input images and compare them visually. Specifically, we show the attention maps of the $1^{st}$, $6^{th}$, and $12^{th}$ self-attention layers in Fig. 1, Fig. 2, and Fig. 3, respectively, using randomly selected 6 input images from the CelebA (Liu et al., 2015) dataset. In these figures, from top to bottom, each row is the attention map of a DiT model trained with $N=10, 10^3, 10^4$, and $10^5$ images. Meanwhile, each column is related to an input image. For a better visualization, we use the logarithm normalization on attention maps before applying a colormap. For the same DiT Block, attention maps of different images demonstrate a similar pattern. Interestingly, we find that the attention maps of a DiT's self-attention layers demonstrate a consistent

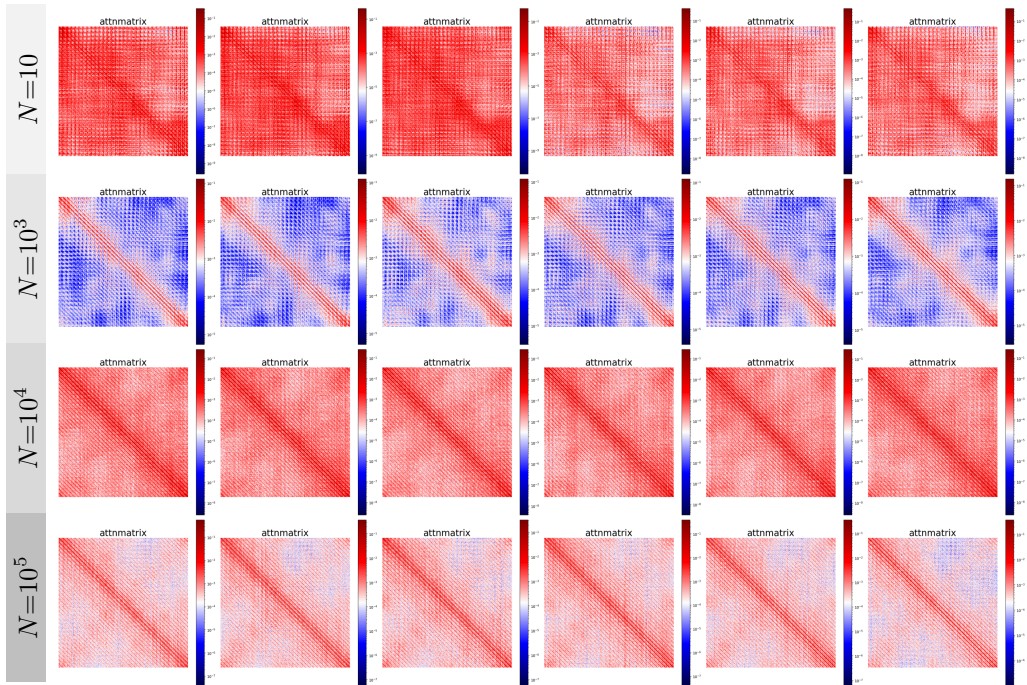

Figure 2: Attention maps of the $6^{\text{th}}$ DiT Block produced by feeding 6 random images to DiT-XS/1 models.

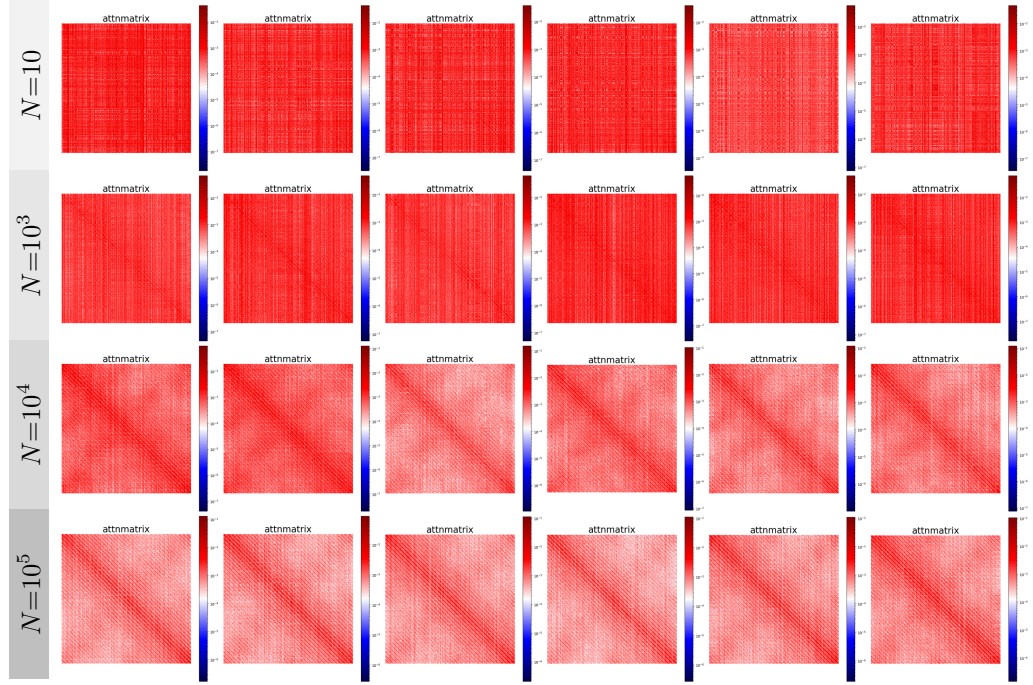

Figure 3: Attention maps of the $12^{\text{th}}$ DiT Block produced by feeding 6 random images to DiT-XS/1 models.

pattern among different input images, suggesting that the attention maps of a DiT, after training, are part of its inductive biases rather than being mostly governed by a specific input image.

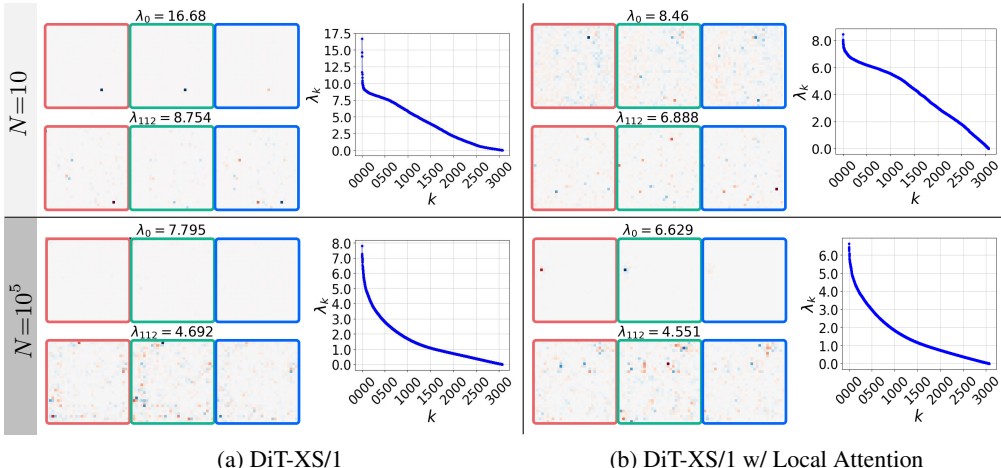

(a) DiT-XS/1           (b) DiT-XS/1 w/ Local Attention

Figure 4: Jacobian eigenvector comparison between DiT-XS/1 w/ and w/o using local attention. In both cases, their Jacobian eigenvectors do not exhibit the harmonic bases observed in a UNet.

## C  JACOBIAN EIGENVECTOR ANALYSIS OF USING LOCAL ATTENTION

The geometry-adaptive harmonic bases extracted via Jacobian eigenvectors is the inductive bias that drives the generalization of UNet-based diffusion models. Our analysis shows that these harmonic bases do not exist in a DiT. To further verify our finding, we extract the Jacobian eigenvectors of a DiT equipped with local attention. Fig. 4 compares the Jacobian eigenvectors of a DiT-XS/1 with and without using local attention. We follow Kadkhodaie et al. (2024) to extract the Jacobian eigenvectors and perform the analysis discussed in the main paper. The Jacobian eigenvectors of both DiTs demonstrate similar sparse patterns, showing no harmonic bases similar to the one observed in simplified (Kadkhodaie et al., 2024) and normal UNets. This observation corroborates our finding that harmonic bases are not the driving factor of DiT's generalization, regardless of the employed attention type.

## D  ADDITIONAL VISUALIZATIONS OF DIT'S ATTENTION MAPS

In Fig. 4 of the main paper, we visualize the attention maps of a DiT trained with $N=10^3$, $10^4$, and $10^5$ images, applying a colormap to the interval $[0, 0.1]$. More specifically, we first normalize an attention map to the range of $[0, 1.0]$. Then we apply the colormap to attention maps with an upper bound of $0.1$, meaning that all values larger than $0.1$ are colored identically. We choose $0.1$ as the upper bound to make sure patterns of attention maps are easy to read. To further demonstrate the importance of attention map locality for the generalization of a DiT, we use different upper bounds for the colormap. Fig. 5 and Fig. 6 show attention maps with colormap upper bound $0.3$ and $0.5$, respectively. The stronger attention map locality can still be observed when increasing training image number $N$ in both figures, confirming that attention map locality is an inductive bias of a DiT rather than being caused by a specific colormap upper bound.

## E  ADDITIONAL QUANTITATIVE RESULTS

We present more quantitative results using an additional dataset (MSCOCO (Lin et al., 2014)), UNet, extra DiT backbones, and latent-space diffusion models.

### E.1  MORE QUANTITATIVE RESULTS WITH PIXEL-SPACE DIFFUSION MODELS

To confirm our findings in the main paper that attention map locality is an inductive bias that drives the generalization of a DiT, we present more quantitative comparisons. We provide the PSNR gap results in Tab. 2 and the FID in Tab. 3.

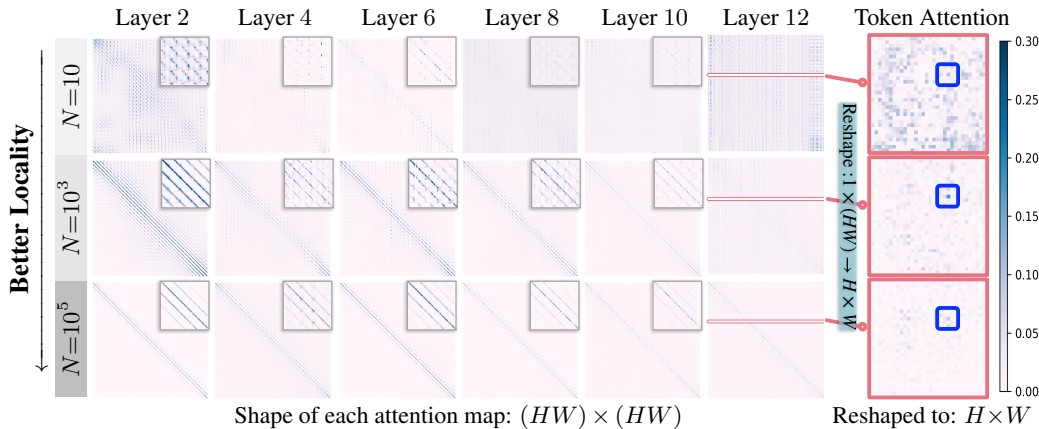

Figure 5: Attention maps of DiTs trained with $10$, $10^3$, and $10^5$ images. All attention maps are linearly normalized to the range $[0, 1]$, with a colormap applied to the interval $[0, 0.3]$.

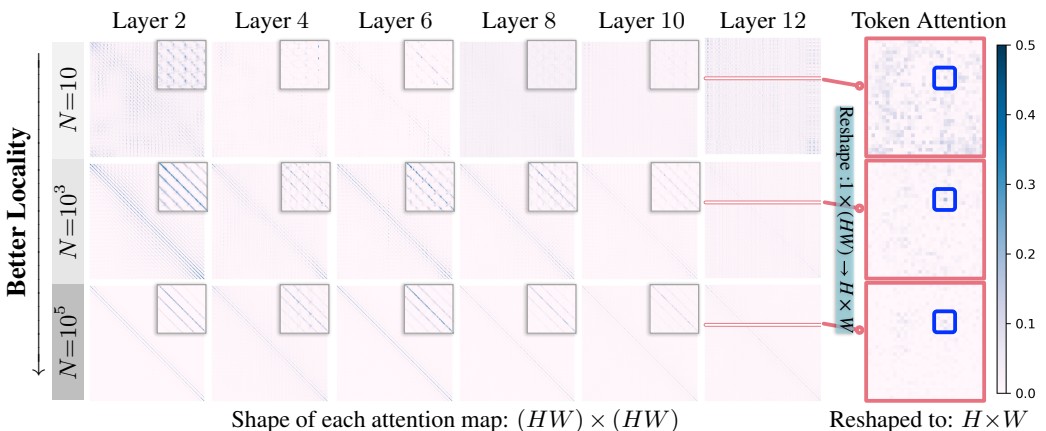

Figure 6: Attention maps of DiTs trained with $10$, $10^3$, and $10^5$ images. All attention maps are linearly normalized to the range $[0, 1]$, with a colormap applied to the interval $[0, 0.5]$.

In the main paper, we find that a UNet has a worse generalization ability than a DiT with the same FLOPs, when measured by the PSNR gap. Tab. 2 and Tab. 3 demonstrate that the PSNR gap and FID of a UNet are worse than those of a DiT when the training image number $N$ is insufficient and are better than those of a DiT when the training image number $N$ is sufficient. This observation aligns well with our findings in the main paper.

Reducing the complexity of a neural network is a well-known way to improve a model's generalization when the dataset is small. In Tab. 2 and Tab. 3, we compare the PSNR gap and FID of DiT-XS/1 and DiT-XXS/1. The latter is a smaller model with fewer hidden dimensions. A smaller DiT can reduce the PSNR gap and the FID when the training image number $N$ is small. We recognize reducing network complexity as an orthogonal way to improve a DiT's generalization. However, importantly, Tab. 2 and Tab. 3 show that it is less effective than using local attention. We present an additional study to reducing a DiT's parameters in Appendix G.

MSCOCO (Lin et al., 2014) is a dataset where the long-range correspondences might prompt a DiT to learn more global attention maps. To assess this, we compare the PSNR gap and FID between UNet, DiT, and DiT with local attention using MSCOCO. We find that using local attention can still reduce a DiT's PSNR gap and FID score, confirming that the attention map locality is an important inductive bias for a DiT's generalization, even for a dataset where long-range correspondences are important.

Table 2: PSNR gap↓ comparison based on pixel-space diffusion model. The training images have a resolution of $32\times32$.

| Model | CelebA | | | ImageNet | | | MSCOCO | | |
|---|---|---|---|---|---|---|---|---|---|
| | $N=10^3$ | $N=10^4$ | $N=10^5$ | $N=10^3$ | $N=10^4$ | $N=10^5$ | $N=10^3$ | $N=10^4$ | $N=10^5$ |
| UNet | 13.86 | 5.53 | 0.06 | 13.39 | 4.84 | 0.05 | 13.65 | 5.20 | 0.13 |
| DiT-XXS/1 | 7.40 | 0.71 | 0.01 | 7.13 | 0.43 | 0.05 | 7.40 | 0.52 | 0.13 |
| DiT-XS/1 | 7.49 | 0.80 | 0.01 | 7.77 | 1.08 | 0.05 | 7.36 | 0.60 | 0.13 |
| DiT-XS/1 w/ Local | 6.56 | 0.57 | 0.01 | 6.76 | 0.74 | 0.05 | 6.36 | 0.41 | 0.13 |

Table 3: FID↓ comparison based on a pixel-space diffusion model. The training images have a resolution of $32\times32$.

| Model | CelebA | | ImageNet | | MSCOCO | |
|---|---|---|---|---|---|---|
| | $N=10^4$ | $N=10^5$ | $N=10^4$ | $N=10^5$ | $N=10^4$ | $N=10^5$ |
| UNet | 9.8136 | 3.3871 | 61.3965 | 13.1302 | 58.4580 | 7.0214 |
| DiT-XXS/1 | 9.0085 | 2.5749 | 33.2946 | 20.3075 | 26.0462 | 13.6076 |
| DiT-XS/1 | 9.6932 | 2.6303 | 52.5650 | 17.3114 | 28.3496 | 12.9695 |
| DiT-XS/1 w/ Local | 8.4258 | 2.4988 | 43.8687 | 18.0671 | 24.4308 | 13.4735 |

## E.2 QUANTITATIVE RESULTS WITH LATENT DIFFUSION MODEL

The attention map locality is identified to be the inductive bias that drives the generalization of a pixel-space DiT. In addition, we study whether the latent-space DiT also demonstrates such an inductive bias. To clarify, we use the pre-trained VAE from the official repository of DiT (Peebles & Xie, 2023). Then we train a DiT on the latent space of the pre-trained VAE, where the training images have a shape of $256\times256$ with the corresponding latent code being of resolution $32\times32$.

Tab. 4 presents the PSNR gap and FID results comparing UNet, DiT-XS/1 and DiT-XS/1 with local attention, using CelebA and MSCOCO data. Comparing UNet and DiT-XS/1, DiT-XS/1 has a smaller PSNR gap and FID when the training image number $N$ is small, reconfirming the observation of the pixel-space experiments.

Comparing DiT-XS/1 with and without local attention, we observe the use of local attention to reduce the PSNR gap. However, we do not observe a smaller FID value when $N$ is small, making it different from the pixel-space DiT. To investigate this further, we compare the attention map between pixel-space and latent-space DiTs in Fig. 7. We observe that the attention map locality gap between $N=10^3$ and $N=10^5$ is larger in pixel-space DiT than in latent-space DiT. We speculate that this is because larger training images ($256\times256$ for the latent DiT compared with $32\times32$ for pixel DiT), coupled with the VAE encoder, create more diverse information that enables a latent DiT to more easily achieve good generalization (reflected by attention map locality). Because of this, it is hard to improve a DiT's FID further when $N$ is small.

## F QUALITATIVE RESULTS

In addition to the quantitative comparison, we present qualitative results with LSUN Bridge, LSUN Church, and ImageNet datasets in Fig. 8. With and without using local attention, two DiTs generate images of similar quality. Taking a closer look, for some samples, we find that DiTs trained with $10^4$ and $10^5$ images while using local attention produces images that are more like each other than DiTs trained without using local attention. For example, in each subfigure, the samples highlighted with green boxes are more similar, irrespective of whether the model was trained with $N=10^4$ and $10=10^5$ images, than the samples surrounded with red boxes. This phenomenon aligns with our finding that the use of local attention can improve a DiT's generalization.

To further verify the above observation we verify that the use of local attention makes images generated by a DiT trained with $10^4$ images more similar to images generated by a DiT trained with

Table 4: PSNR gap↓ and FID↓ comparison based on latent diffusion model with CelebA dataset. All models are trained with $256\times256$ images, where the corresponding latent codes have a resolution of $32\times32$.

| Model | CelebA | | | | | MSCOCO | | | | |
| --- | --- | --- | --- | --- | --- | --- | --- | --- | --- | --- |
| | PSNR Gap | | | FID | | PSNR Gap | | | FID | |
| | $N{=}10^3$ | $N{=}10^4$ | $N{=}10^5$ | $N{=}10^4$ | $N{=}10^5$ | $N{=}10^3$ | $N{=}10^4$ | $N{=}10^5$ | $N{=}10^4$ | $N{=}10^5$ |
| UNet | 8.69 | 3.35 | 0.11 | 36.5862 | 6.4805 | 5.74 | 1.84 | 0.12 | 159.7947 | 41.9908 |
| DiT-XS/1 | 3.36 | 1.19 | 0.07 | 12.7045 | 8.5751 | 2.08 | 0.14 | 0.11 | 72.3063 | 68.1251 |
| DiT-XS/1 w/ Local | 2.21 | 0.17 | 0.07 | 13.6513 | 9.2621 | 1.19 | 0.13 | 0.10 | 78.8400 | 74.1681 |

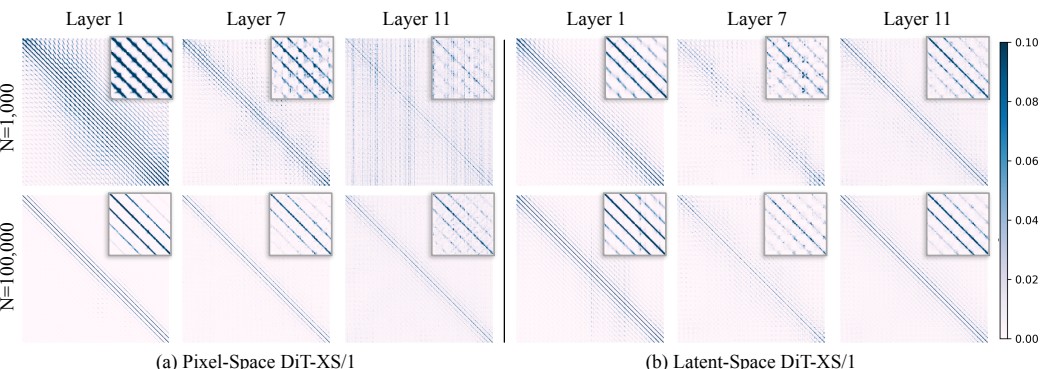

(a) Pixel-Space DiT-XS/1          (b) Latent-Space DiT-XS/1

Figure 7: Attention map comparison between pixel-space and latent-space DiTs. The attention maps of latent-space DiT demonstrate smaller gaps between $N{=}10^3$ and $N{=}10^5$ in terms of the attention map locality.

$10^5$ images. For this we randomly sample $50,000$ images with each model, using a fixed random noise, so that the two models generate the same image content. Then we compute the average pixel intensity difference between the two generations. We show the results on LSUN Church, LSUN Bridge, and ImageNet data in Tab. 5.

## G  REDUCING PARAMETERS OF A DIT

We study two parameter reduction approches for a DiT: parameter sharing and composing attention maps with PCA.

**Parameter Sharing.**  For this approach, the 2[nd] and 4[th] DiT Blocks reuse the parameters of the 1[st] and 3[rd] DiT Blocks, respectively, leading to a DiT model with the same FLOPs but fewer parameters. Fig. 9 (row 3) shows the PSNR and PSNR gap of the DiT using this parameter sharing approach. Meanwhile, Tab. 6 demonstrates the FID of the same model trained with $N{=}10^4$ and $10^5$ images. Notably, sharing parameters in a DiT can reduce the PSNR gap, resulting in FID improvement when $N{=}10^4$. However, it is not as effective as using local attention in modifying the generalization of a DiT.

**Composing Attention Maps with PCA.**  Another parameter reduction approach we explore is composition of attention maps of a DiT with PCA. To elaborate, we first collect the attention maps of 2048 images. Particularly, we noise each image with 10, 25 and 40 steps and obtain the attention maps of all attention heads corresponding to these noisy images, where the sampling diffusion step is set to 50. Taking the DiT-XS/1 model as an example, we collect a total of $24,576 = 2048\,(\text{images})\times3\,(\text{diffusion steps})\times4\,(\text{attention heads})$ attention maps. We use the DiT-XS/1 model trained with the ImageNet (Deng et al., 2009) dataset and collect attention maps of a DiT's first three self-attention layers using randomly selected 2048 images from the testing set of the

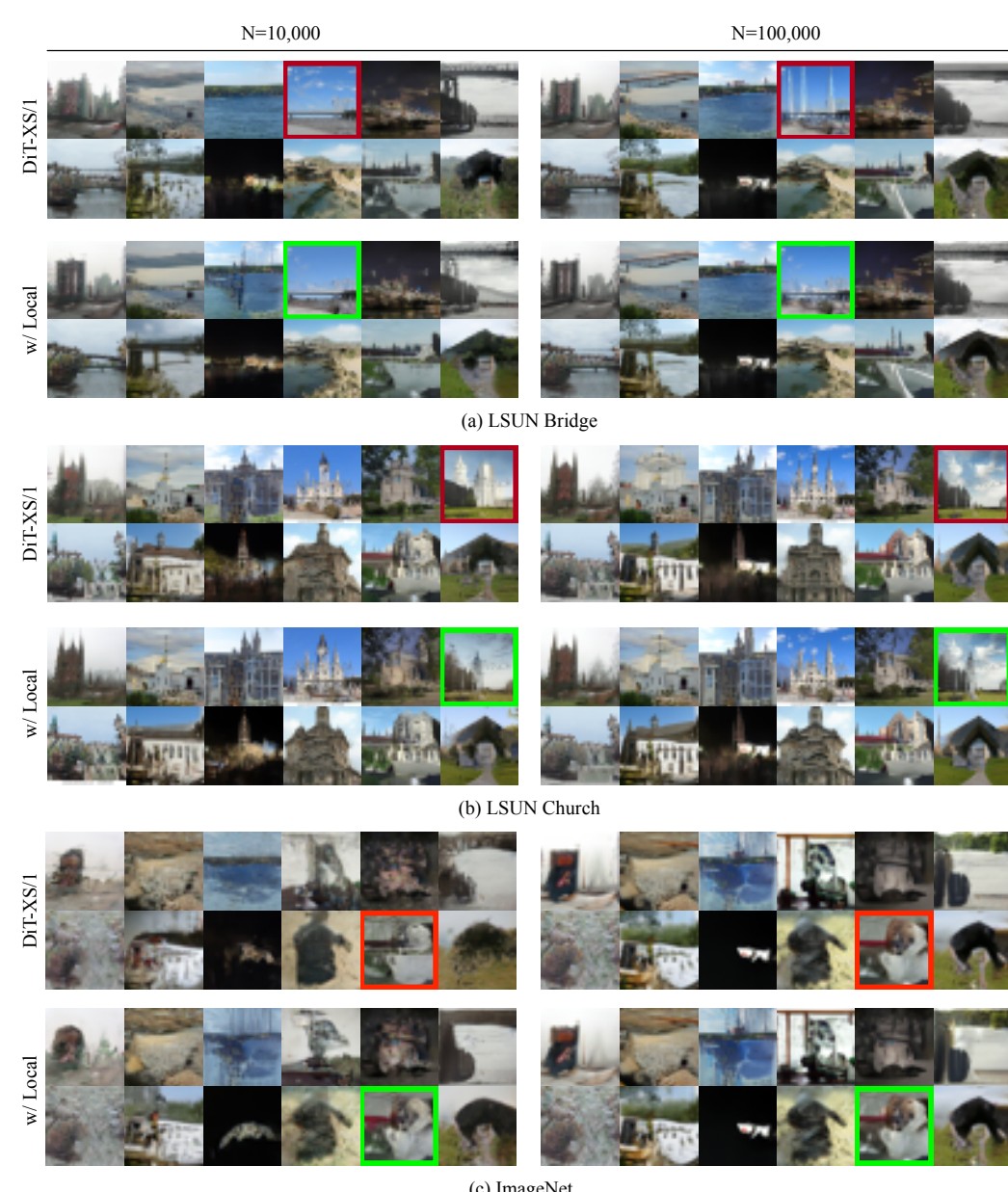

Figure 8: Visual comparison between DiT-XS/1 and DiT-XS/1 w/ Local Attention. As highlighted by red and green boxes, using local attention results in images from models trained with $N=10^4$ and $N=10^5$ images to be closer to each other.

same dataset. Next, we compute the principal components of each self-attention layer from the corresponding attention maps. We use the low rank PCA function[‡] of PyTorch and obtain the first 50 principal components, where each principal component has the same size as the attention map. Fig. 10 shows the principal components and the corresponding coefficients for the first three DiT Blocks. Notably, we find that PCA is effective in capturing the dominant diagonal patterns that indicate the locality in a DiT's attention map. Finally, we use the principal components of the attention maps to reduce the parameters of a DiT. Concretely, we replace the two MLP layers that map the input tensor to query and key matrices by a smaller MLP mapping the input tensor to 50 coefficients for each

---

[‡]https://pytorch.org/docs/stable/generated/torch.pca_lowrank.html

Table 5: Averaged pixel intensity difference between generations of models trained with $N{=}10^4$ and $N{=}10^5$ images using pixel-space DiT-XS/1 with and without local attention. The generated images have a resolution of $32{\times}32$. Using local attention reduces the averaged pixel intensity difference.

| Model | LSUN Bridge | LSUN Church | ImageNet |
|---|---|---|---|
| DiT-XS/1 | 13.0078 | 12.9620 | 17.5844 |
| DiT-XS/1 w/ Local | 11.0645 | 10.9402 | 15.2034 |

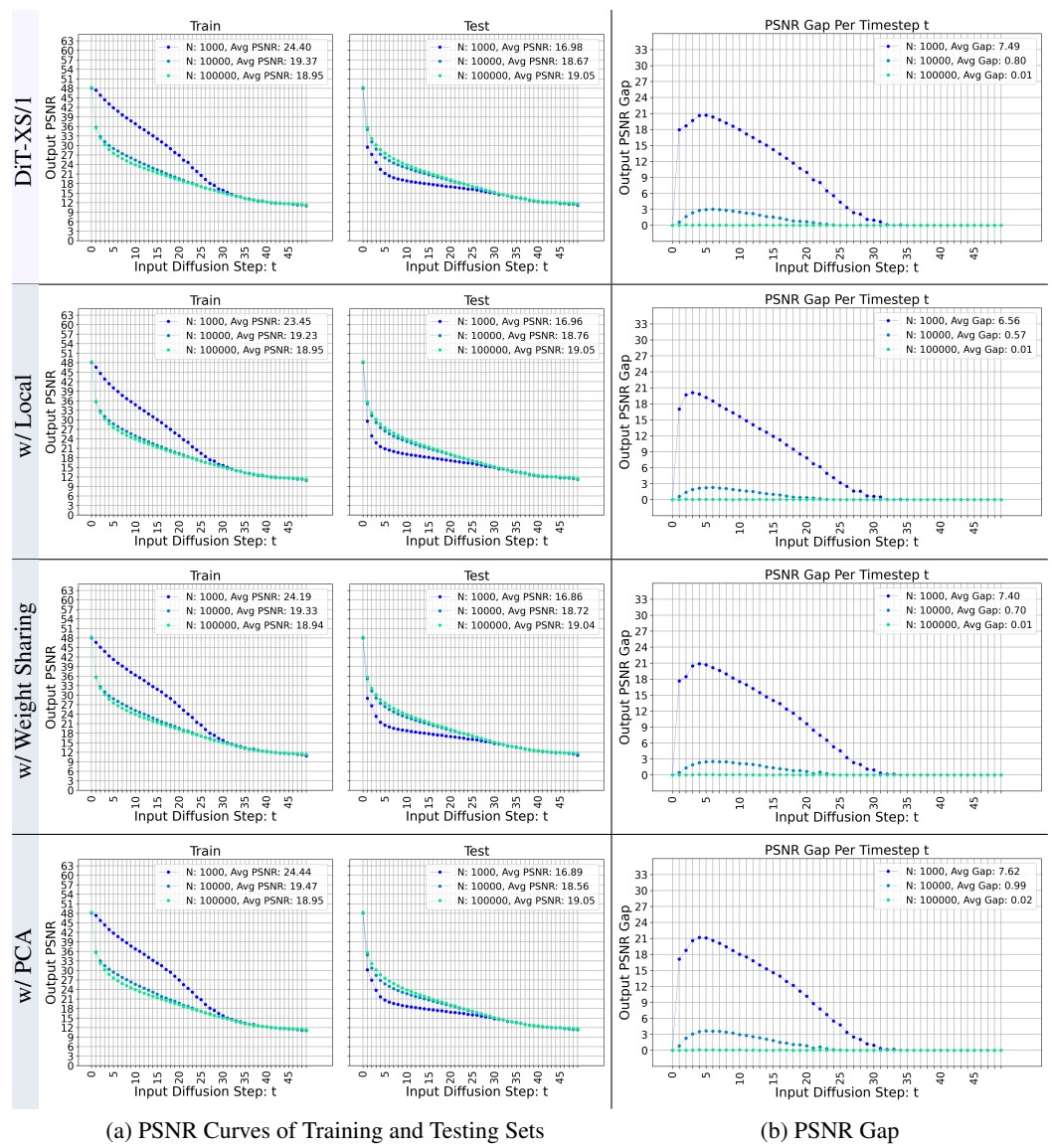

(a) PSNR Curves of Training and Testing Sets      (b) PSNR Gap

Figure 9: The PSNR (a) and PSNR gap (b) comparison. Taking DiT-XS/1 as the baseline (row 1), using local attention (row 2) can achieve decent PSNR gap improvement. In contrast, using parameter reduction approaches: weight sharing (row 3) and PCA (row 4), hardly achieve a PSNR gap improvement (row 3) or even make it worse then the baseline (row 4).

principal component (PC). Then the new attention map is obtained as follows:

$$\text{Attention Map} = \text{Coefficients} \odot \text{PCs} + \delta, \tag{1}$$

Table 6: FID↓ comparison between reducing DiT FLOPs and parameter size. Using parameter sharing is not as effective in reducing FID as the local attention. Using PCA will make the FID worse when $N=10^4$. Both are not as effective as using local attention in improving the FID with $10^4$ training images.

| Model | CelebA | |
|---|---|---|
| | $N=10^4$ | $N=10^5$ |
| DiT-XS/1 | 9.6932 | 2.6303 |
| w/ Local | 8.4580 | 2.5469 |
| | $-12.74\%$ | $-3.17\%$ |
| w/ Weight Sharing | 8.7819 | 2.5802 |
| | $-9.40\%$ | $-1.90\%$ |
| w/ PCA | 11.3872 | 2.5482 |
| | $+17.48\%$ | $-3.12\%$ |

where $\odot$ denotes matrix multiplication while $\delta = \dfrac{1.0}{1024}$ is used to ensure that the attention weights for a specific token sum to 1. We replace the normal attention maps of a DiT's first three self-attention layers with the attention maps composed with principal components following Eq. (1). According to the PSNR and PSNR gap comparison in Fig. 9 (row 4) as well as the FID comparison in Tab. 6, reducing parameters of DiT by composing its first three attention maps with principal components cannot reduce a DiT's PSNR gap, leading to a worse FID when $N=10^4$.

## H  CONNECTION TO THEORETICAL RESULTS

In this section we'll provide connections to theoretical work (De Wolf, 2008; Yang & Salman, 2019; Vasudeva et al., 2024) that can be used to explain our empirical findings.

We will start by discussing preliminaries from prior work on the simplicity bias of transformers (Vasudeva et al., 2024) in Appendix H.1. We'll subsequently connect this work to our results in Appendix H.2 and show that local attention encourages the low sensitivity bias of a transformer. Finally, in Appendix H.3, we demonstrate low sensitivity of a transformer is connected to better generalization.

### H.1  PRELIMINARIES

Prior work (Vasudeva et al., 2024) showed that attention modules learn simpler features more quickly, which implies that the transformer is biased towards simple functions and lower sensitivity. To show this, Vasudeva et al. (2024) considered a model with at least one self-attention layer. To simplify the analysis, prior work removes the non-linear Softmax function from a standard self-attention layer and focuses on linear attention of the form

$$\Phi = \frac{\boldsymbol{x}W_q \cdot W_k^\top \boldsymbol{x}^\top}{\sqrt{\dim}} \cdot \boldsymbol{x}W_v, \tag{2}$$

with input $\boldsymbol{x} \in \mathbb{R}^{T \times \tilde{d}}$ and dim the scaling dimension of the attention layer. Further, $W_q$, $W_k$, and $W_v$ are trainable parameters that map the input $\boldsymbol{x}$ to query, key, and value, respectively. Below, we use $d = T\tilde{d}$.

Under the assumption that a transformer with linear self-attention layers works in a boolean space $\{0,1\}^d$, Vasudeva et al. (2024) showed the following main results: A transformer model $f(\boldsymbol{x})$ that contains at least one self-attention layer can be represented by the linear combination of a set of orthonormal monomial terms

$$f(\boldsymbol{x}) = \sum_{U \subseteq [d]} \hat{f}(U)\chi_U(\boldsymbol{x}), \quad \chi_U := \prod_{i \in U} x_i, \quad \forall U \subseteq [d], \tag{3}$$

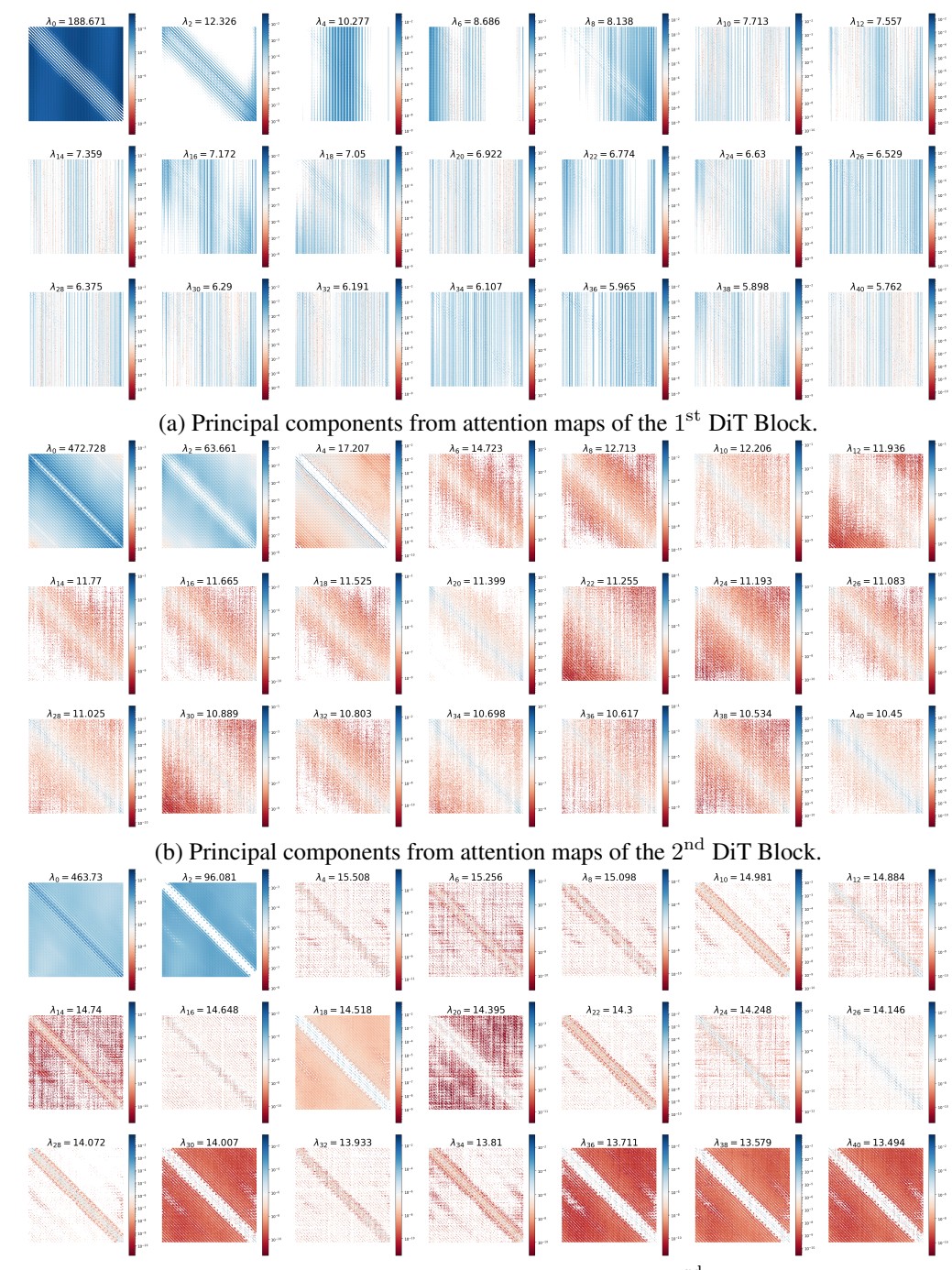

(a) Principal components from attention maps of the $1^{\text{st}}$ DiT Block.

(b) Principal components from attention maps of the $2^{\text{nd}}$ DiT Block.

(c) Principal components from attention maps of the $3^{\text{rd}}$ DiT Block.

Figure 10: Principal components extracted from attention maps of different DiT Blocks. Based on a DiT-XS/1 model trained with $N=10^5$ data from ImageNet, we perform PCA on attention maps of its first three layers, using 2048 images, resulting in a total of 24576 attention maps. For a better visualization, we adopt the logarithm normalization to principal components before applying colormaps.

where set $U \subseteq [d]=\{1,\dots,d\}$, and term $\hat{f}(U)$ is the coefficient for a monomial term. For an input sequence $x$, these orthonormal monomial terms, under their specific assumptions (Yang & Salman, 2019), form a set of Fourier bases (De Wolf, 2008).

In addition, at an input location $\boldsymbol{x}$, Vasudeva et al. (2024) compute the eigenvalues of those orthonormal monomial terms $\chi_U$, which form an eigenfunction, via

$$\mu_{|U|} := \mathop{\mathbb{E}}_{\boldsymbol{x} \sim \{0,1\}^d} \left[ \chi_U K(\boldsymbol{x}, \mathbf{1}) \right]. \tag{4}$$

In Eq. (4), $\mu_{|U|}$ is the eigenvalue for monomial $\chi_U$, $|U|$ denotes the size of $U$, $\mathbf{1}$ is a vector of all ones in space $\{0,1\}^d$, and $K(\cdot)$ represents a neural kernel (Yang & Salman, 2019; Hron et al., 2020), *e.g.*, conjugate kernel (CK) or neural tangent kernel (NTK). Based on theorems discussed in prior work (Yang & Salman, 2019; Hron et al., 2020), Vasudeva et al. (2024) theoretically prove that eigenvalues $\mu_{|U|}$ for $U \subseteq [d]$ satisfy

$$\begin{aligned} \mu_0 \geq \mu_2 \geq \cdots \geq \mu_{2k} \geq \ldots, \\ \mu_1 \geq \mu_3 \geq \cdots \geq \mu_{2k+1} \geq \ldots. \end{aligned} \tag{5}$$

This result is important because it explains why attention modules learn simpler features more quickly: the eigenvalues of monomial terms with lower degree are larger as shown in Eq. (5). This indicates that transformers are biased towards polynomials with lower orders. Considering that a low-degree polynomial tends to have low sensitivity, this result also implies that transformers are biased toward low sensitivity functions.

## H.2 RELATION TO INDUCTIVE BIASES IN DIFFUSION TRANSFORMERS

This result is relevant because it provides a theoretical foundation for our work. Concretely, when using global attention, $U \subseteq [d]$ is not restricted in any form. This hence means that any elements in the input tensor $\boldsymbol{x} \in \mathbb{R}^{T \times \bar{d}}$ can interact with each other.

In contrast, using local attention restricts the interaction between elements in the input tensor as illustrated in Figs. 4 and 5 of our main paper. This implies that $U$ now only represents a subset of the possible interactions, which reduces the order of the highest degree monomial $\chi_U$ significantly.

Because the highest degree monomials are of much lower order, local attention lowers the sensitivity of the transformer *w.r.t.* data perturbations.

## H.3 LOWER SENSITIVITY LEADS TO BETTER GENERALIZATION

Under the linear self-attention assumption, Vasudeva et al. (2024) also demonstrate that sensitivity of a transformer of data perturbation is connected with the sharpness of the minima, *i.e.*, the sensitivity of the loss for small changes of the network weight near minima of the parameter space. The low sharpness of the minima is a widely accepted indicator of model generalization (Keskar et al., 2016; Neyshabur et al., 2017; Jiang et al., 2019) and has been empirically verified for transformers (Hahn & Rofin, 2024). Considering a linear model $\Phi(\theta; \boldsymbol{x}) = \theta^\top \boldsymbol{x}$, where $\theta$ is the weight of the linear layer and $\boldsymbol{x}$ is the input data. Adding a small perturbation $\Delta \boldsymbol{x}$ to input $\boldsymbol{x}$ is equivalent to perturbing the layer weight $\theta$

$$\Phi(\boldsymbol{\theta}; \boldsymbol{x} + \Delta \boldsymbol{x}) = \boldsymbol{\theta}^\top (\boldsymbol{x} + \Delta \boldsymbol{x}) = \Phi(\boldsymbol{\theta}; \boldsymbol{x}) + \boldsymbol{\theta}^\top \Delta \boldsymbol{x} = \Phi(\boldsymbol{\theta}; \boldsymbol{x}) + \Delta \boldsymbol{\theta}^\top \boldsymbol{x} = \Phi(\boldsymbol{\theta} + \Delta \boldsymbol{\theta}; \boldsymbol{x}), \tag{6}$$

where $\Delta \theta = \dfrac{\theta^\top \Delta \boldsymbol{x}}{\|\boldsymbol{x}\|_2^2} \boldsymbol{x}$.

For a more complex model like a transformer, Vasudeva et al. (2024) empirically verified that the connection between the low sensitivity and flat minima still holds. Taking both Eq. (5) and Eq. (6) into consideration, we can draw the conclusion that using local attention reduces the sensitivity of a transformer, resulting in flatter minima, which leads to improved generalization.

# I RAW PSNR AND PSNR GAP CURVES

We show the raw PSNR and PSNR gap curves corresponding to models in Fig. 6 of the main manuscript in Fig. 11 to Fig. 16. Besides the PSNR gap change after using local attentions as discussed in Sec. 3.1 of the main manuscript, given sufficient images ($N = 10^5$) of a specific dataset, the training PSNR curves and the average PSNR are very similar between a DiT with and without using local attentions, suggesting that using local attention can mostly maintain a DiT's dataset fitting ability. This is likely because all the important information of a DiT's attention map is retained.

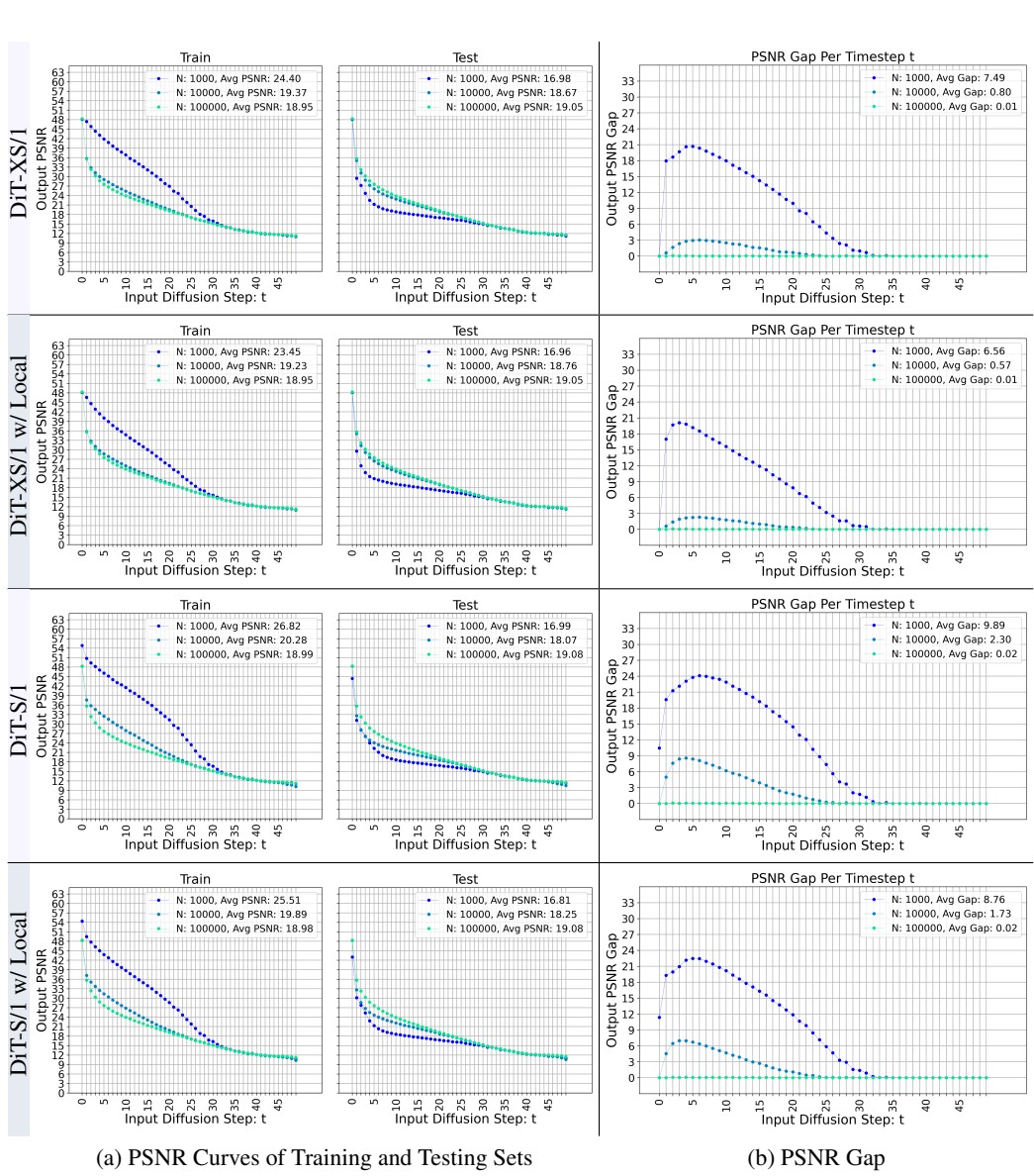

(a) PSNR Curves of Training and Testing Sets          (b) PSNR Gap

Figure 11: The PSNR (a) and PSNR gap (b) comparison on the CelebA (Liu et al., 2015) dataset.

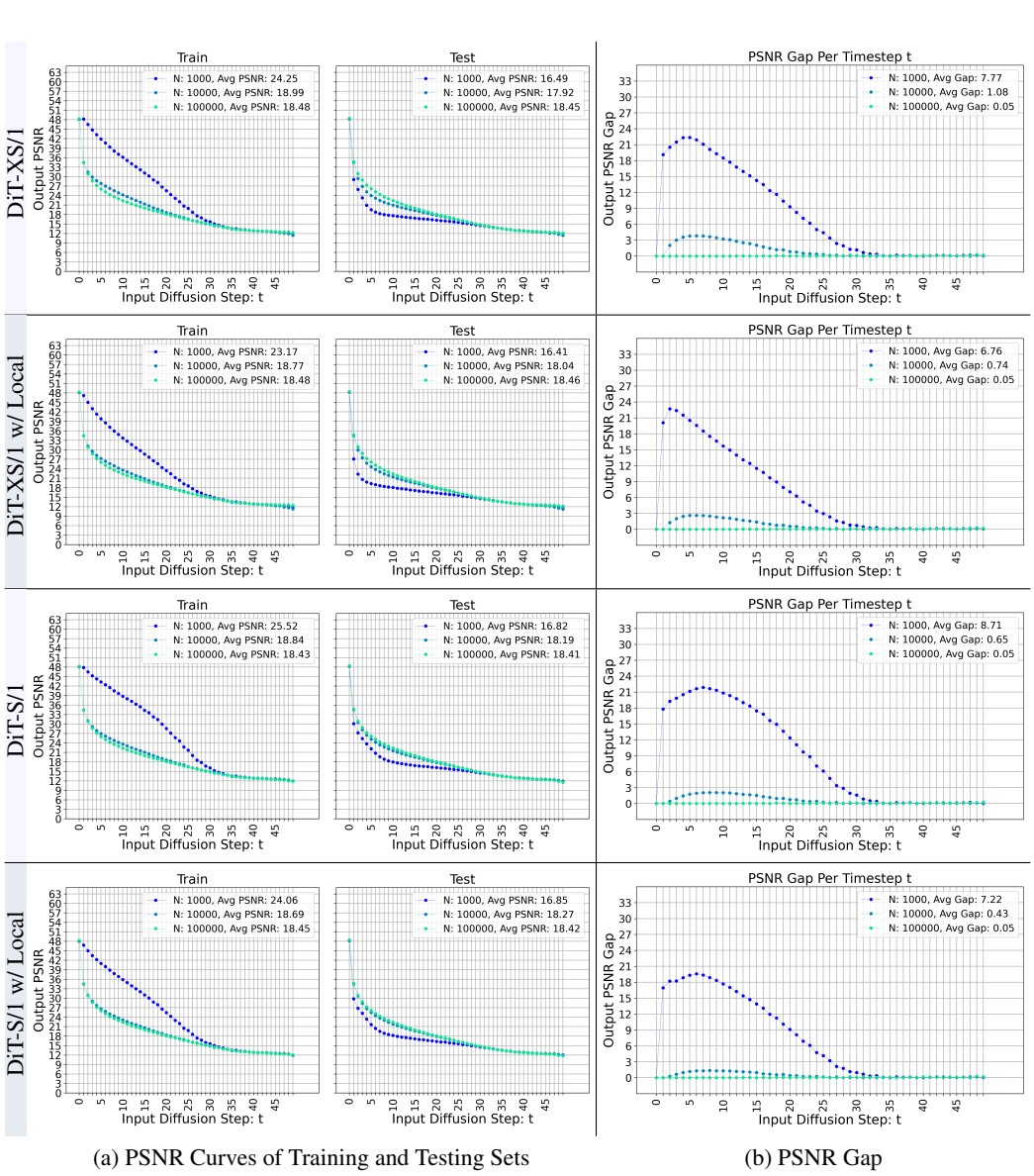

(a) PSNR Curves of Training and Testing Sets          (b) PSNR Gap

Figure 12: The PSNR (a) and PSNR gap (b) comparison on the ImageNet (Deng et al., 2009) dataset.

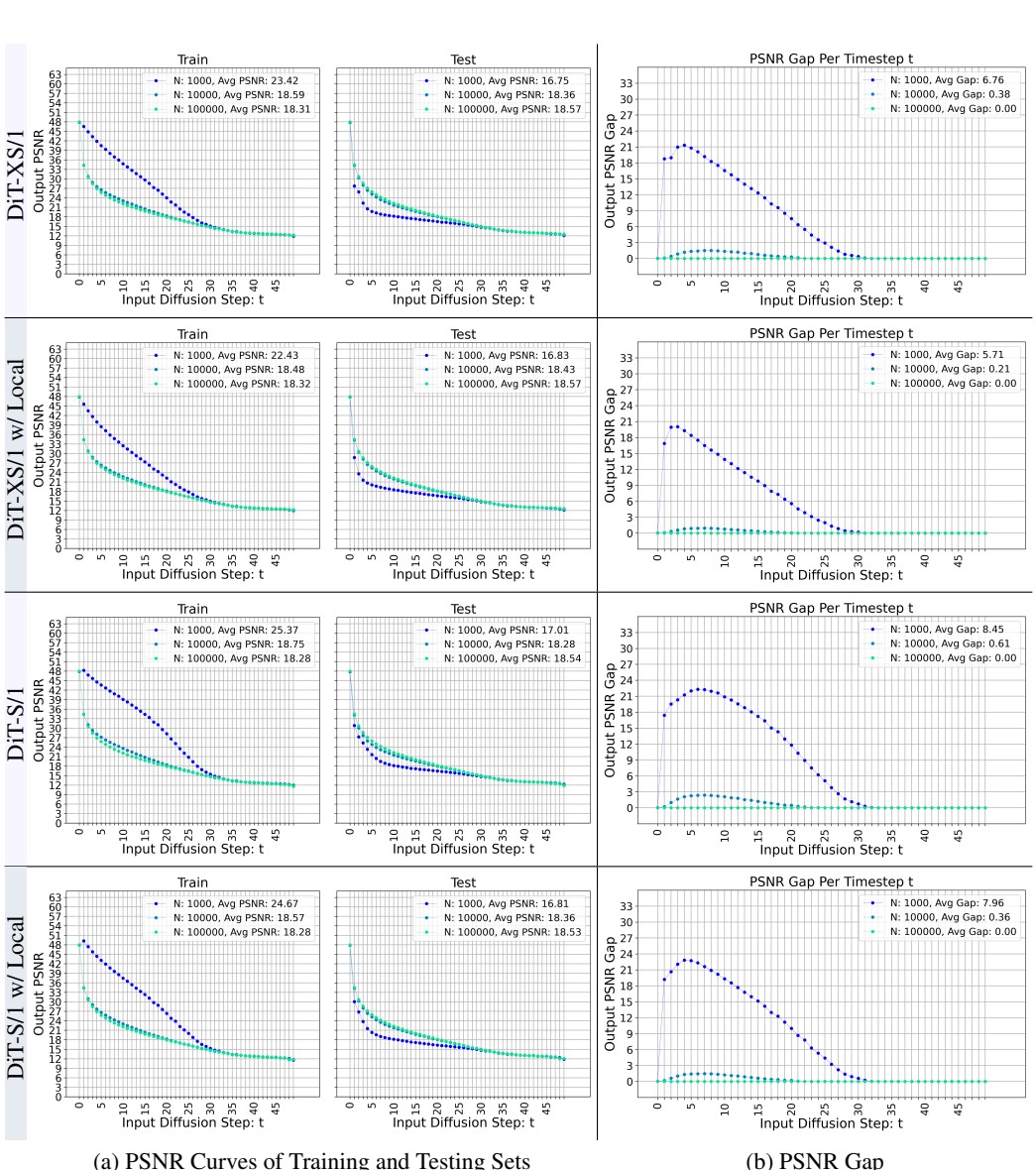

(a) PSNR Curves of Training and Testing Sets    (b) PSNR Gap

Figure 13: The PSNR (a) and PSNR gap (b) comparison on the LSUN (Church) (Yu et al., 2015) dataset.

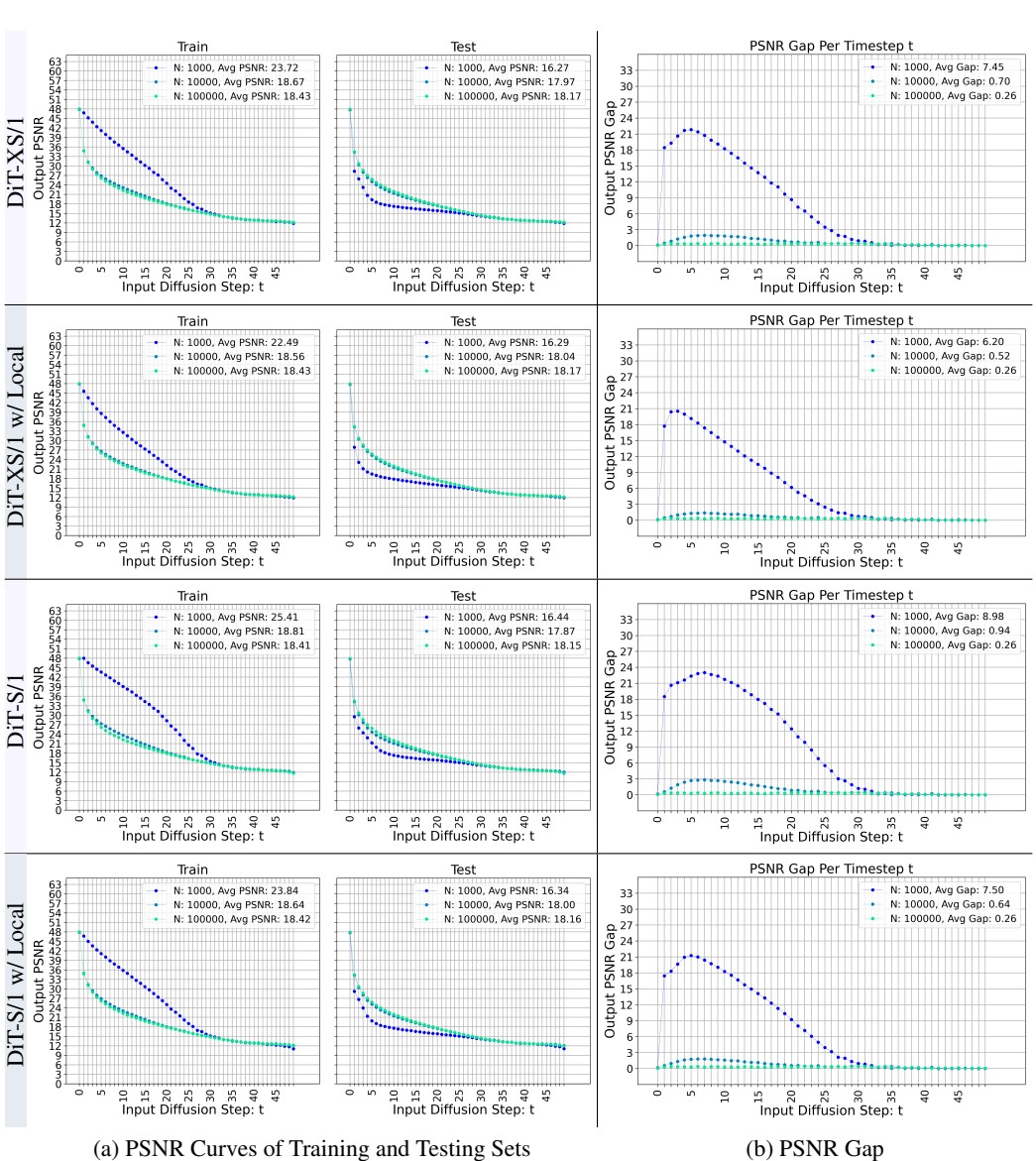

(a) PSNR Curves of Training and Testing Sets         (b) PSNR Gap

Figure 14: The PSNR (a) and PSNR gap (b) comparison on the LSUN (Bedroom) (Yu et al., 2015) dataset.

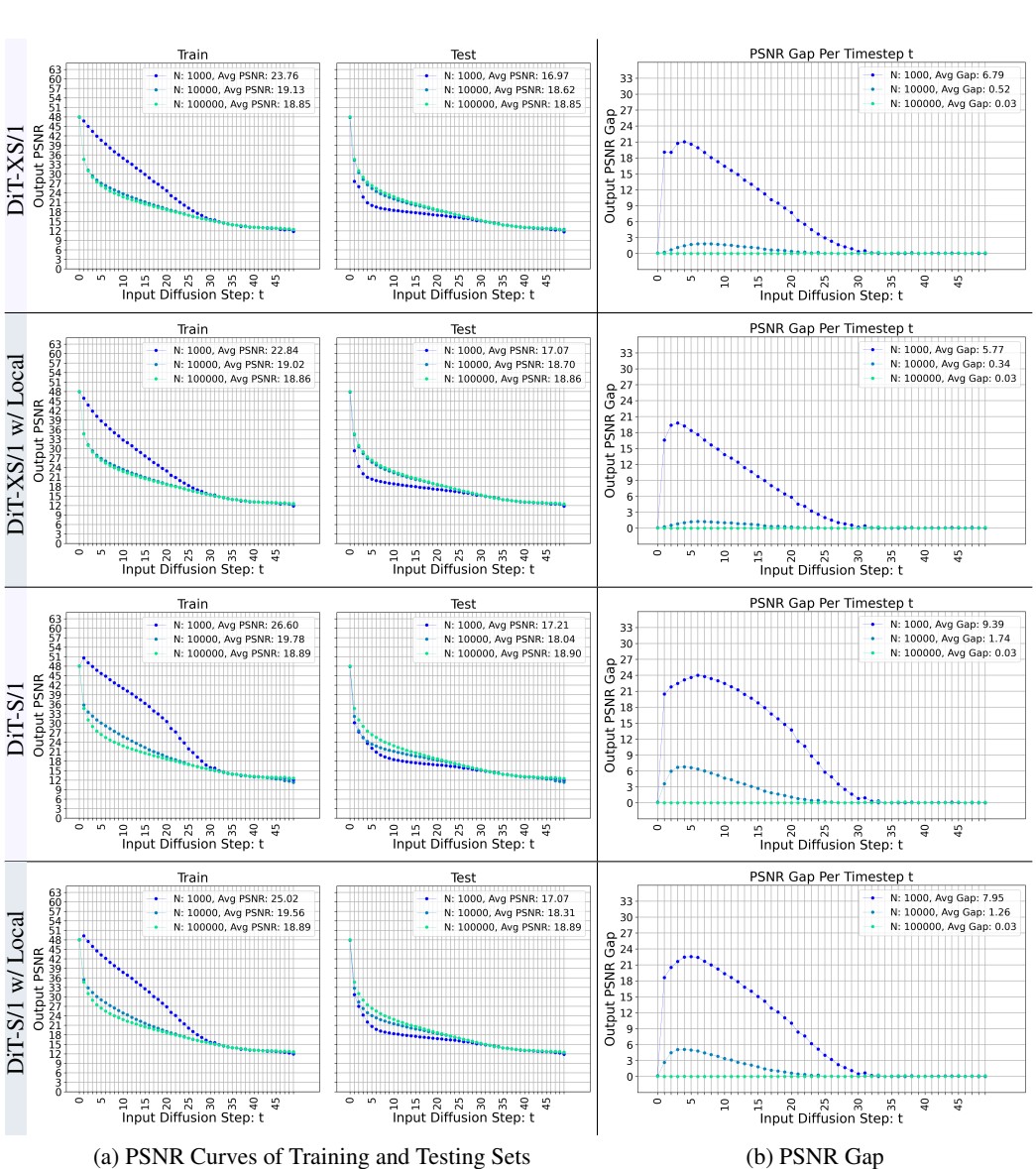

(a) PSNR Curves of Training and Testing Sets      (b) PSNR Gap

Figure 15: The PSNR (a) and PSNR gap (b) comparison on the LSUN (Bridge) (Yu et al., 2015) dataset.

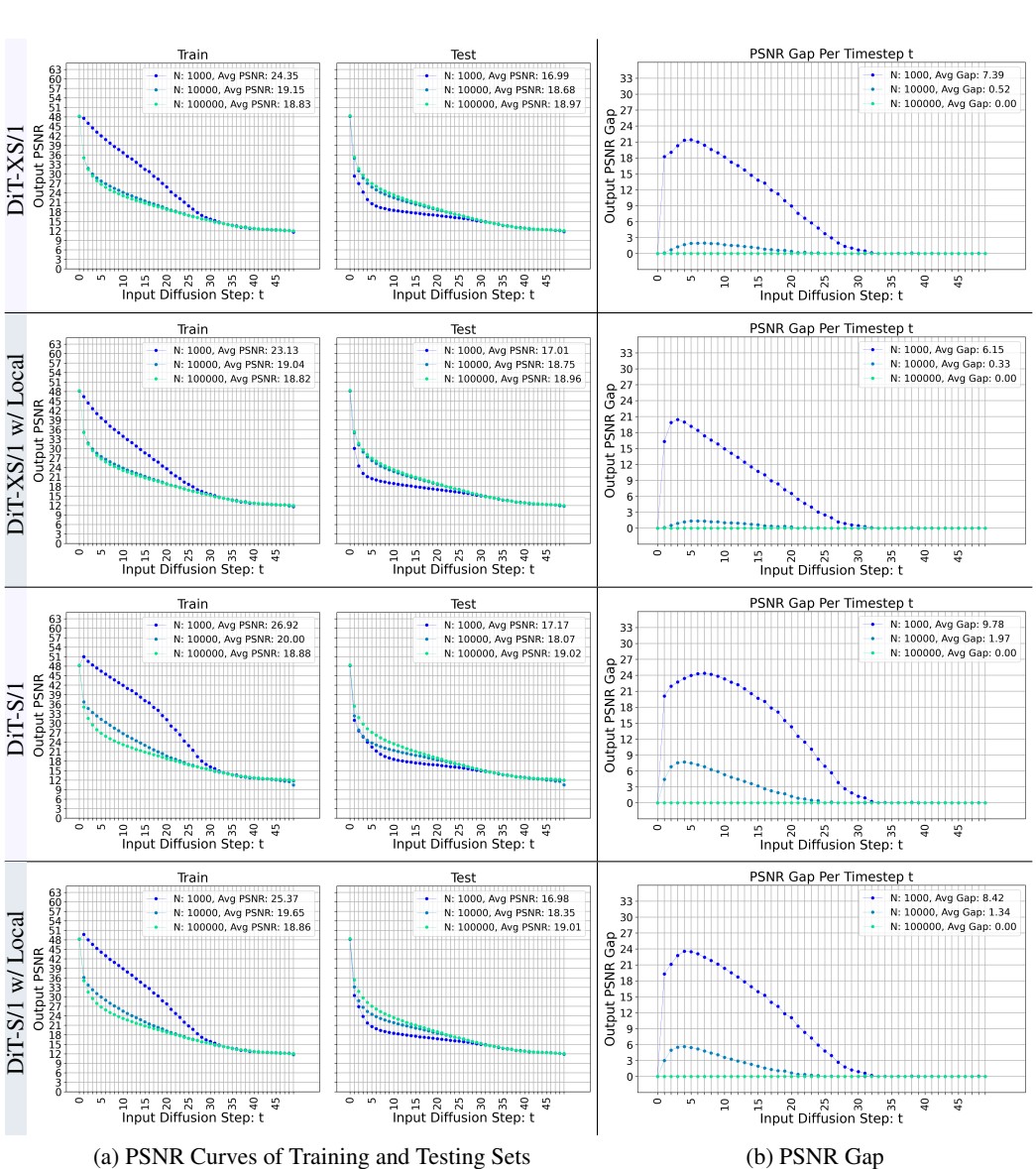

(a) PSNR Curves of Training and Testing Sets      (b) PSNR Gap

Figure 16: The PSNR (a) and PSNR gap (b) comparison on the LSUN (Tower) (Yu et al., 2015) dataset.