# OpenReview forum: "On Inductive Biases That Enable Generalization in Diffusion Transformers"
_ICLR.cc/2025/Conference — Submitted to ICLR 2025_

### Official Review · Reviewer_MCon · 2024-10-28

**Soundness:** 2
**Presentation:** 2
**Contribution:** 2
**Rating:** 3
**Confidence:** 3

**Summary:**

This paper investigates the generalization mechanisms of diffusion models, comparing the inductive biases of UNet-based and Transformer-based diffusion models (DiTs). The authors find that while geometry-adaptive harmonic bases drive generalization in UNets, DiTs rely on the locality of attention maps. This paper modifies attention window sizes to enhance DiT generalization, especially in settings with limited training data.

**Strengths:**

This paper considers the locality of attention maps as the key inductive bias that contributes to the generalization of DiTs, which is an interesting perspective. In this way, the authors control generalization of DiTs by local attention window.

**Weaknesses:**

1.	Findings in this paper need to be further verified. Although Fig. 6 shows that using local attentions in a DiT can improve its generalization (measured by PSNR gap) with limited training data conditions, using a pruned DiTs may also improve its generalization ability. It is a well-known generalization phenomenon that an appropriate ratio between model size/complexity and data quantity achieves a better performance. In this way, decreasing the model size/complexity may be the key reason for improving generalization ability with limited training data conditions, where local attentions is just one way for decreasing model size/complexity.

2.	The contribution is limited. The authors introduce local attention windows to improve the generalization of DiTs, which have been widely used in previous studies [cite1-3]. Furthermore, the authors do not provide a theoretical explanation for why local attention windows improve generalization in DiTs. Therefore, the contribution of this paper is limited.

[cite1] Chenglin Yang, Siyuan Qiao, Qihang Yu, Xiaoding Yuan, Yukun Zhu, Alan Yuille, Hartwig Adam, and Liang-Chieh Chen. Moat: Alternating mobile convolution and attention brings strong vision models. In ICLR, 2022
[cite2] Ali Hatamizadeh, Greg Heinrich, Hongxu Yin, Andrew Tao, Jose M Alvarez, Jan Kautz, and Pavlo Molchanov. Fastervit: Fast vision transformers with hierarchical attention. arXiv preprint arXiv:2306.06189, 2023
[cite3] Ali Hassani, StevenWalton, Jiachen Li, Shen Li, and Humphrey Shi. Neighborhood attention transformer. In CVPR, 2023

3.	The experimental results are not convincing. The paper relies heavily on FID scores to evaluate the performance of DiTs. Expanding the evaluation to other relevant metrics, such as CLIP Score [cite4-6], could provide a more comprehensive assessment of the model's capabilities.

[cite4] Alec Radford, Jong Wook Kim, Chris Hallacy, Aditya Ramesh, Gabriel Goh, Sandhini Agarwal, Girish Sastry, Amanda Askell, Pamela Mishkin, Jack Clark, et al. Learning transferable visual models from natural language supervision. In International conference on machine learning, pp. 8748–8763. PMLR, 2021.
[cite 5] Richard Zhang, Phillip Isola, Alexei A Efros, Eli Shechtman, and Oliver Wang. The unreasonable effectiveness of deep features as a perceptual metric. In Proceedings of the IEEE Conference on Computer Vision and Pattern Recognition, pp. 586–595, 2018.
[cite6] Alain Hore and Djemel Ziou. Image quality metrics: PSNR vs. SSIM. In 20th International Conference on Pattern Recognition, pp. 2366–2369. IEEE, 2010.

4.	The improvements shown in the paper focus on limited training data scenarios, but the impact of the proposed method on other datasets or high-resolution tasks is not thoroughly explored, e.g., MS COCO and PartiPrompts.
5.	I suggest the authors provide more visual comparisons to demonstrate the generalization improvement of DiTs.

**Questions:**

Please see the Weaknesses.

---

> ### Author Response · Authors · 2024-11-23
>
> * **Q: The reduction of the model complexity is the key factor of the improved generalization. Local attention is just a way to reduce a model’s complexity.**
>
>   **A**: We agree that local attention reduces the complexity of a DiT. But reducing the model complexity does **NOT** always lead to the improved generalization. For example, in Fig. 7 (line 444) and Tab. 2 (line 433) of the main paper, we use the same local attention in different layers (head, mix, tail) of a DiT. While all these models have the same complexity, their generalization differs significantly. Particularly, the tail setting makes the generalization worse.
>
>   | FID  | $N{=}10^4$ | $N{=}10^5$ |
>   | --- | --- | --- |
>   | DiT-XS/1 (CelebA) | 9.6932 |  2.6303 |
>   | DiT-XS/1 w/ Local (head) (CelebA) | 8.4580 |  2.5469 |
>   | DiT-XS/1 w/ Local (tail) (CelebA) | 18.0717 |  2.4288 |
>   | DiT-XS/1 (ImageNet) | 52.5650 |  17.3114 |
>   | DiT-XS/1 w/ Local (head) (ImageNet) | 43.8687 |  18.0671 |
>   | DiT-XS/1 w/ Local (tail) (ImageNet) | 59.8510 |  17.5818 |
>
>   In addition, in Tab. 6 (line 540) of Appendix G, we study the effects of reducing a DiT’s parameters using weight sharing and PCA. We have added a new experiment in Tab. 2 (line 324) and Tab. 3 (line 335) of Appendix E, reducing the attention dimension of a DiT. Those experiments show mixed results and none of them are more effective than using local attention.
>
>   | FID (CelebA) | $N{=}10^4$ | $N{=}10^5$ |
>   | --- | --- | --- |
>   | DiT-XS/1 | 9.6932 |  2.6303 |
>   | DiT-XS/1 w/ Local | 8.4580 |  2.5469 |
>   | DiT-XS/1 w/ Weight Sharing | 8.7819 |  2.5802 |
>   | DiT-XS/1 w/ PCA | 11.3872 |  2.5482 |
>   | DiT-XXS/1 | 9.0085 |  2.5749 |
>
>   Our inductive bias analysis of DiT reveals local attention as an effective way to reduce the complexity of a DiT.
> ---
>
> * **Q: The contribution of this paper is limited since local attention has been widely used and no theoretical analysis is presented.**
>
>   **A**: We are **NOT** trying to find **novel ways to improve generalization** of a DiT. Instead, we focus on studying the **inductive bias** that drives the generalization of a DiT. Through a Jacobian eigenvector analysis, we first show that DiT’s generalization is not driven by harmonic bases. Then we analyze DiT’s attention maps and reveal the attention map locality as the inductive bias of a DiT. Finally, we add local attention to a DiT to modify its generalization, as our target is to **verify** that attention locality is truly the inductive bias of DiT.
>
>   Indeed, local attention has been widely used. However, prior use focuses on model efficiency. To our best knowledge, no prior work studies the use of local attention in order to modify the generalization of a DiT. A theoretical analysis of the generalization of modern DiTs and UNets has been a challenge, mainly due to the complex operations used by these networks. To make the theoretical analysis possible, Kadkhodaie et al. [1] use a one-channel simplified UNet, which differs significantly from widely used UNets, leaving a gap between the theoretical analysis and networks used in real world applications. This work focuses more on an empirical analysis of normal DiTs and UNets, providing insights that can be applied to modern diffusion models more directly, which we believe is equally important.
> ---
>
> * **Q: Expanding the evaluation to other relevant metrics, such as CLIP Score.**
>
>   **A**: We use the PSNR gap to measure the generalization of a diffusion model, and FID to measure the generation quality. Since all our experiments are unconditional, without paired images or text, metrics such as CLIP score are not applicable.

---

> ### Author Response · Authors · 2024-11-23
>
> * **Q: Experimental results on MSCOCO and PartiPrompts.**
>
>   **A**: We include experimental results using MSCOCO data in Tab. 2 (PSNR gap, line 324) and Tab. 3 (FID, line 335) of the revised Appendix E.1.
>
>   For the PSNR gap comparison using MSCOCO data we obtain:
>
>   | PSNR Gap (MSCOCO)  | $N{=}10^3$ | $N{=}10^4$ | $N{=}10^5$ |
>   | --- | --- | --- | --- |
>   | DiT-XS/1  | 7.36 | 0.60 | 0.13 |
>   | DiT-XS/1 w/ Local | 6.36 | 0.41 | 0.13 |
>
>   Using local attention reduces the PSNR gap when $N{=}10^3$ and $N{=}10^4$, which aligns well with our observations on other datasets shown in Tab. 1 of the main paper.
>
>   For the FID comparison using MSCOCO data we obtain:
>
>   | FID (MSCOCO)  | $N{=}10^4$ | $N{=}10^5$ |
>   | --- | --- | --- |
>   | DiT-XS/1  | 28.3496 | 12.9695 |
>   | DiT-XS/1 w/ Local | 24.4308 | 13.4735 |
>
>   Using local attention reduces the FID value when the number of training images is small, which also aligns with the results on other datasets as shown in Fig. 6 of the main paper.
>
>   PartiPrompts benchmark only contains text prompts and has no images. All of our experiments are based on unconditional image generation, making PartiPrompts not applicable.
> ---
>
> * **Q: Qualitative results.**
>
>   **A**: We included a qualitative comparison between DiT-XS/1 with and without local attention in Fig. 8 (line 472) of the revised Appendix F. For some samples, we find that DiTs trained with $10^4$ and $10^5$ images while using local attention produce results that are more like each other than DiTs trained without using local attention. We compute the averaged pixel intensity difference between DiTs trained with $10^4$ and $10^5$ images in Tab. 5 (line 486) of the revised Appendix F, which quantitatively verifies the above observation.
>
>   | Avg Pixel Diff | LSUN Bridge | LSUN Church | ImageNet|
>   | --- | --- | --- | --- |
>   | DiT-XS/1|  13.0078|  12.9620|  17.5844|
>   | DiT-XS/1 w/ Local | 11.0645|  10.9402 | 15.2034|
> ---
> [1] *Zahra Kadkhodaie, Florentin Guth, Eero P Simoncelli, and Stéphane Mallat. Generalization in diffusion models arises from geometry-adaptive harmonic representation. In ICLR, 2024*

---

> > ### Author Response · Authors · 2024-11-26
> >
> > Thank you for your valuable comments that help us improve this paper. We hope our response as well as the revised paper and supplementary material are able to answer your questions. We are looking forward to hearing back from you and answering any new questions that you may have. Thank you again for your effort to improve this paper!

---

> > > ### Comment · Reviewer_MCon · 2024-11-26
> > >
> > > Thanks for your response. Concerns 1, 2, and 4 have not been solved.
> > >
> > > **Concern 1**: According to the experimental results in Table 6, the reduction of the model complexity is the key factor of the improved generalization. Local attention is just a way to reduce a model’s complexity.
> > >
> > > **Concern 2**: Local attention windows have been widely used in previous studies [cite1-3]. Therefore, a theoretical explanation for why local attention windows improve generalization in DiTs is indeed important.
> > >
> > > **Concern 4**: The impact of the proposed method needs to be evaluated on PartiPrompts.
> > >
> > > So I am keeping the rating.

---

> > > > ### Author Response · Authors · 2024-11-26
> > > >
> > > > Thanks for your time and feedback. Please note that **this is an analysis paper rather than a methodology paper**. Our main contributions are two **insights**: 1) a DiT doesn’t exhibit the same inductive bias as a diffusion UNet; 2) instead, locality of attention is an inductive bias of a DiT that drives its generalization and arises during training. Said differently, we can assess generalization of a trained DiT model via its attention locality, as illustrated in Fig. 2 (line 230). Note, the focus of our paper is **not** to improve the generalization of a DiT via a local attention operation. Instead, a local attention operation is only used to verify correctness of the discovered inductive bias.
> > > >
> > > > * **Q: Evaluation on PartiPrompts.**
> > > >
> > > >   **A**: **PartiPrompts data is not applicable because PartiPrompts data is a set of text prompts.** Note, in our paper we study inductive biases for **unconditional image generation**, i.e., we train and evaluate using solely **images**. No text prompts are used in any of our models or results.
> > > > ---
> > > >
> > > > * **Q: According to Table 6, reduction of the model complexity is the key factor of the improved generalization. Local attention is just a way to reduce a model’s complexity.**
> > > >
> > > >   **A**: Locality of attention (as opposed to the local attention operation) arises during training (see Fig. 2), and is not just a way to reduce model complexity. Tab. 6 (and also Fig. 2) verifies that locality of attention is indeed an inductive bias of a well trained DiT. Concretely, enforcing locality of attention improves FID by 12.7% when training with a smaller dataset. In contrast, it isn’t as effective to use parameter sharing or a PCA, which both have about the same FLOP count as a model where locality of attention is enforced. These results suggest that naively reducing the model complexity does not always modify generalization. It is hence important to understand ways that drive generalization, which is the motivation of our paper.
> > > > ---
> > > >
> > > > * **Q: A theoretical explanation for why local attention windows improve generalization in DiTs is indeed important.**
> > > >
> > > >   **A**: A theoretical explanation showing why locality of attention improves generalization is undoubtedly valuable. However, providing rigorous theoretical statements for complex deep nets like a DiT, especially trained on real-world image data, remains an open problem in our community. Expecting such an analysis is beyond the scope of this work.
> > > >
> > > >   Prior work [1] approached this problem using a simplified UNet trained on synthetic data with regular intensity variations and contours, while removing the diffusion noise scheduler. This deviates significantly from real-world diffusion models. While inspiring, practical relevance is limited. In contrast, our empirical analysis focuses on standard diffusion architectures and realistic data, reducing the gap between theory and practice. We think this is also impactful.
> > > > ---
> > > > [1] *Zahra Kadkhodaie, Florentin Guth, Eero P Simoncelli, and Stéphane Mallat. Generalization in diffusion models arises from geometry-adaptive harmonic representation. In ICLR, 2024*

---

> ### Comment · Reviewer_MCon · 2024-11-27
>
> Thanks for your response. I agree that **naively reducing the model complexity does not always modify generalization.** Therefore, local attention is just a way to reduce a model’s complexity, which can improve generalization.  I also agree that **this is an analysis paper rather than a methodology paper.** Therefore, **a theoretical explanation for why local attention windows improve generalization in DiTs is indeed important for an analysis paper.**

---

> > ### Author Response · Authors · 2024-11-28
> >
> > Thank you for your valuable feedback and the active participation in the discussion. We particularly appreciate your suggestion about a theoretical explanation of our empirical findings, which we think make our paper more comprehensive.
> >
> > We have uploaded a new revision, adding `Appendix H`, where we provide connections to theoretical work [2,3,4] that can be used to explain our empirical findings.
> >
> > Our theoretical explanation is based on prior work about the simplicity bias of transformers [2]. Preliminaries are introduced in Appendix H.1. Subsequently, we connect prior work [2] to our results in Appendix H.2 and show that local attention can encourage the simplicity bias, resulting in low sensitivity *w.r.t.* data perturbation of a transformer. Finally in Appendix H.3, we demonstrate that low sensitivity of a transformer is connected with the existence of flat minima, which is a widely accepted indicator of good generalization of a model.
> >
> > Again we sincerely thank you for the valuable suggestion.
> >
> > ---
> > [2] *Bhavya Vasudeva, Deqing Fu, Tianyi Zhou, Elliott Kau, Youqi Huang, and Vatsal Sharan. Simplicity bias of transformers to learn low sensitivity functions. arXiv preprint arXiv:2403.06925, 2024.*
> >
> > [3] *Ronald De Wolf. A brief introduction to fourier analysis on the boolean cube. Theory of Computing, 2008.*
> >
> > [4] *Greg Yang and Hadi Salman. A fine-grained spectral perspective on neural networks. arXiv preprint arXiv:1907.10599, 2019.*

---

> > > ### Comment · Reviewer_MCon · 2024-11-29
> > >
> > > Thank you for your response. After carefully reading the revised paper and response, l am still confused about Appendix H.2. Specifically, what's the relation between the sensitivity of the transformer and the inductive bias? The current version of the theoretical explanation is not rigorous enough and is unclear. These theoretical explanations are very important for an analysis paper, and should be moved into the main text, instead of the Appendix.

---

> ### Author Response · Authors · 2024-12-01
>
> Thank you for the active participation in the discussion. We address the questions below.
>
> * **Q: Theoretical explanations should be moved into the main text.**
>
>   **A**: Thank you for the suggestion. We followed prior work on this topic [1] in keeping the empirical analysis in the main paper, while deferring the theoretical analysis to the appendix. We are happy to move the theoretical explanation from Appendix H to the main paper if all reviewers and AC concur.

---

> > ### Author Response · Authors · 2024-12-01
> >
> > * **Q: Relation between sensitivity and inductive bias.**
> >
> >   **A**: We appreciate the reviewer’s effort in thoroughly checking the provided theoretical explanation. Thanks a lot. It would help us if the reviewer could pinpoint which connection is not clear to the reviewer such that we can focus on explaining the specifics. As we don’t exactly know what the reviewer aims to understand better, we next go through Appendix H line-by-line and explain each statement below after highlighting the most important terms:
> >
> >   * `Term Explanation`:
> >     * **Simplicity Bias** [9,10,11,12]: The inductive bias of a neural network that encourages learning of “simple” functions.
> >
> >     * **Spectral Simplicity Bias** [13]: A specific type of simplicity bias. Based on the polynomial representation of a neural network, this inductive bias asserts that a neural network tends to learn low order monomial terms faster than high order monomial terms.
> >
> >     * **Low Sensitivity Bias** [14]: The inductive bias of a neural network (particularly transformer) that encourages learning of low sensitivity functions.
> >
> >     * **Attention Locality Bias**: The inductive bias of a transformer we discovered through our empirical analysis, i.e., the fact that well trained attention exhibits locality. Our theoretical explanation connects this inductive bias to the above (spectral) simplicity bias and low sensitivity bias introduced by prior work, as we explain next.
> >
> >   * `line 588-593`: Under the assumption that a deep net $f(x)$ contains at least one linear attention layer in a space of real-valued functions on the boolean cube {$0,1$}$^d$, $f(x)$ can be represented by a linear combination of a set of monomial terms $\chi_U$.
> >
> >   * `line 593`: $\chi_U:=\prod_{i\in U}x_i, \forall U\subseteq [d]$. $x_i$ is the $i^{\rm th}$ element of the input. The function $\chi_U$ is a multilinear monomial consisting of input elements $x_i$, where $i \in U$. A deep net $f(x)$ can be represented by a polynomial composed of the linear combination of monomials $\chi_U, U\subseteq [d]$.
> >
> >   * `line 646-647`: Using the Fourier analysis approach to a neural network [3] (Sec. 2.2), these monomial terms $\chi_U$ form a set of Fourier bases.
> >
> >   * `line 648-651`: According to [2] and [4], one can obtain the eigenvalue $\mu_{\|U\|}$ corresponding to each monomial term $\chi_U$, with the help of neural kernels [4].
> >
> >   * `line 654-659`: According to [2] (Proposition 3.1) and [4] (Theorem 4.1), with ascending set size $|U|$, the magnitudes of the eigenvalues $\mu_{\|U\|}$ corresponding to monomial terms $\chi_U$ decrease, splitted by even and odd terms. The proof is in Appendix B of [2].
> >
> >   * `line 661-662`: The larger eigenvalue of a lower order monomial term indicates that the transformer is biased towards polynomials with lower orders, demonstrating that a transformer has a **(spectral) simplicity bias** [2,4].
> >
> >   * `line 662-664`: A lower order polynomial has a low sensitivity, *i.e.*, it reacts slowly when the input has a random change. Thus, according to [14], the **(spectral) simplicity bias** leads to a **low sensitivity bias**.
> >
> >   * `line 668-670`: Global attention in a transformer does not restrict the set $U\subseteq [d]:=${$1,...,d$}, meaning the corresponding polynomial takes into account all possible high order terms.
> >
> >   * `line 671-673`: **Attention locality** restricts the size of set $U$, as also illustrated in Figs. 4 and 5 of our main paper. This removes the high order monomials $\chi_U$. Hence, the deep net $f(x)$ is represented by a polynomial composed of a linear combination of fewer and lower order monomials $\chi_U$.
> >
> >   * `line 674-675`: Removing high degree monomials further encourages the **(spectral) simplicity bias**. It consequently also encourages the **low sensitivity bias**.
> >
> >   * `line 678-682`: For deep nets, the **sharpness of minima** [15] is a widely accepted indicator [5,6,7,8] for a model’s **generalization**.
> >
> >   * `line 683-689`: For a linear model, as shown in Eq. (6), model sensitivity (towards input perturbation) is equivalent to perturbing the model weight, meaning that **low sensitivity** is connected with **flat minima** and hence **improved generalization** [5,6,7,8].
> >
> >   * `line 690-691`: For a transformer, [2] empirically verified that the connection between **low sensitivity** and **flat minima** and hence improved **generalization** [5,6,7,8] still holds.
> >
> >   * `line 691-693`: Consequently, **local attention** leads to stronger **(spectral) simplicity bias** and **low sensitivity bias**. This leads to **flatter minima**, which has been widely accepted as an indicator of a model’s **generalization** [5,6,7,8].
> >
> > We are happy to include all or part of the above discussion into the theoretical explanation according to the reviewer’s suggestion. Suggestions are very much welcome.

---

> > > ### Author Response · Authors · 2024-12-01
> > >
> > > [1] *Zahra Kadkhodaie, Florentin Guth, Eero P Simoncelli, and Stéphane Mallat. Generalization in diffusion models arises from geometry-adaptive harmonic representation. In ICLR, 2024*
> > >
> > > [2] *Bhavya Vasudeva, Deqing Fu, Tianyi Zhou, Elliott Kau, Youqi Huang, and Vatsal Sharan. Simplicity bias of transformers to learn low sensitivity functions. arXiv preprint arXiv:2403.06925, 2024*
> > >
> > > [3] *Ronald De Wolf. A brief introduction to fourier analysis on the boolean cube. Theory of Computing, 2008*
> > >
> > > [4] *Greg Yang and Hadi Salman. A fine-grained spectral perspective on neural networks. arXiv preprint arXiv:1907.10599, 2019*
> > >
> > > [5] *Nitish Shirish Keskar, Dheevatsa Mudigere, Jorge Nocedal, Mikhail Smelyanskiy, and Ping Tak Peter Tang. On large-batch training for deep learning: Generalization gap and sharp minima. arXiv preprint arXiv:1609.04836, 2016*
> > >
> > > [6] *Behnam Neyshabur, Srinadh Bhojanapalli, David McAllester, and Nati Srebro. Exploring generalization in deep learning. NeurIPS, 2017*
> > >
> > > [7] *Yiding Jiang, Behnam Neyshabur, Hossein Mobahi, Dilip Krishnan, and Samy Bengio. Fantastic generalization measures and where to find them. arXiv preprint arXiv:1912.02178, 2019*
> > >
> > > [8] *Michael Hahn and Mark Rofin. Why are sensitive functions hard for transformers? arXiv preprint arXiv:2402.09963, 2024*
> > >
> > > [9] *Huh, M., Mobahi, H., Zhang, R., Cheung, B., Agrawal, P., and Isola, P.. The low-rank simplicity bias in deep networks. TMLR, 2023*
> > >
> > > [10] *Lyu, K., Li, Z., Wang, R., and Arora, S.. Gradient descent on two-layer nets: Margin maximization and simplicity bias. NeurIPS, 2021*
> > >
> > > [11] *Shah, H., Tamuly, K., Raghunathan, A., Jain, P., and Netrapalli, P.. The pitfalls of simplicity bias in neural networks. NeurIPS, 2020*
> > >
> > > [12] *Valle-Perez, G., Camargo, C. Q., and Louis, A. A.. Deep learning generalizes because the parameter-function map is biased towards simple functions. ICLR, 2019*
> > >
> > > [13] *Rahaman, N., Baratin, A., Arpit, D., Draxler, F., Lin, M., Hamprecht, F., Bengio, Y., and Courville, A.. On the spectral bias of neural networks. ICML, 2019*
> > >
> > > [14] *Bhattamishra, S., Patel, A., Kanade, V., and Blunsom, P.. Simplicity bias in transformers and their ability to learn sparse Boolean functions. ACL, 2023*
> > >
> > > [15] *Keskar, N. S., Mudigere, D., Nocedal, J., Smelyanskiy, M., and Tang, P. T. P.. On large-batch training for deep learning: Generalization gap and sharp minima. ICLR, 2017*

---

> > > > ### Author Response · Authors · 2024-12-02
> > > >
> > > > Thank you for your active participation in the discussion process. Since the discussion period ends today, we would like to kindly ask whether we have addressed all of your questions regarding the theoretical explanation of the attention locality. We are happy to address any remaining questions if any. Thank you again and we are looking forward to your reply.

---

### Official Review · Reviewer_pa74 · 2024-11-02

**Soundness:** 3
**Presentation:** 3
**Contribution:** 3
**Rating:** 8
**Confidence:** 2

**Summary:**

The paper studies generalization of transfomer-based diffusion models (DiT). The authors first make a connection to prior work that discovered that UNet-based diffusion models generalise because of the inductive biases that can be expressed via geometry-adaptive harmonic bases and show that the same analysis is not able to explain the generalisation in DiTs. Authors instead propose that generalisation in DiTs emerges because of the "locality" of the self-attention layers. They first confirm this empirically, by looking into the attention masks at different layers and discover strong local patterns (i.e., pixels mainly attend to their neighbouring pixels). Then they also propose to use the local attention mask (by masking out all the pixels that are outside of the neighbourhood) and show that this can lead to better performance for smaller datasets (both in terms of the PSNR gap and FID).

I like the paper overall and enjoyed reading it. Hence I vote for acceptance. I am not (at all) familiar with the related literature on inductive biases in diffusion models, so my confidence is low (2/3).

**Strengths:**

- The problem studied (i.e., generalisation of diffusion models) is interesting and highly-relevant. I cannot comment on the novelty aspect since I'm not deeply familiar with this field
- I find the main "theoretical" analysis in the paper interesting, both how they show that the differences with the existing theory for UNets and also how they explain the generalisation of DiTs via the local patterns in the attention masks
- I find the experimental results sufficient for confirming their main hypothesis on the importance of local attention masks

**Weaknesses:**

- The paper is not really self-contained. I feel like this could be improved with including the background section where main concepts used throughout the paper are explained. For example, the Jacobian of the denoising network is crucial for the parts of the analysis, however it is never properly introduced

**Questions:**

- What exactly is the Jacobian you are talking about in Section 2.2? Is it the derivative of the denoising NN w.r.t. its parameters? If so, how do you deal with its dimensionality, since it has dimension of (HW) x M where HxW are the dimensions of the output/image and M is the number of model parameters, right? Also, it is not a diagonal matrix, so do you compute eigen or singular values?
- In line 308 you state you remove the autoencoder, but how do you then go from H x W to (HW) x d, where d is the token dimension?
- Out of curiosity, what is the purpose of black boxes over faces in Figure 1? You are not considering an inpainting task, or?

---

> ### Author Response · Authors · 2024-11-23
>
> * **Q: Including a background section for concepts like Jacobian eigenvector computation.**
>
>   **A**: Thanks for the valuable suggestion. We moved the details of the Jacobian eigenvector computation from the Appendix to Sec. 2 of the revised main paper. We hope this makes the paper more self-contained.
> ---
>
> * **Q: Details of the Jacobian eigenvector computation.**
>
>   **A**: To compute the Jacobian eigenvectors, we first feed a noisy image ($H{\times}W{\times}3$) into a DiT or a UNet and obtain their Jacobian matrix $\nabla\epsilon_\theta\in\mathbb{R}^{\left(H{\times}W{\times}3\right) {\times} \left(H{\times}W{\times}3\right)}$. Each entry of the Jacobian represents the partial derivative of an output pixel w.r.t. an input pixel. Then we perform the eigendecomposition of the Jacobian and obtain the eigenvectors and eigenvalues. Here Jacobian eigenvectors represent the main directions that output pixels respond to input pixel changes.
> ---
>
> * **Q: Regarding the token dimension while removing autoencoder.**
>
>   **A**: The latent space DiT has a token dimension of $4$, determined by the VAE’s latent space size. For pixel-space DiT, we set the token dimension (input channel size) to $3$ since input images have $3$ channels. No other changes were made.
> ---
>
> * **Q: Purpose of face image blackout.**
>
>   **A**: We blackout eyes to avoid privacy issues. In our analysis and experiments, we use the original dataset without blackout.

---

> > ### Comment · Reviewer_pa74 · 2024-11-26
> >
> > Thanks for your answers. I acknowledge I have read the rebuttal as well as the reviews from other reviewers. The main concern from other reviewers seems to be limited novelty/significance of the submitted manuscript, however, I do not share the same concern and find the authors rebuttal convincing enough. So I am keeping my suggestion for paper acceptance (with a low confidence since I am not up to date with the most recent developments in this sub-field).

---

> > > ### Author Response · Authors · 2024-11-26
> > >
> > > Thank you very much for your constructive comments which helped to improve the paper. We really appreciate your support of our paper and the novel insights for generative modeling that are discussed.

---

### Official Review · Reviewer_oYGR · 2024-11-03

**Soundness:** 3
**Presentation:** 2
**Contribution:** 2
**Rating:** 6
**Confidence:** 3

**Summary:**

In this work, the authors identified that an inductive bias of DiT lies in the locality of attention which contributes to the generalizability of DiT. The local attention windows are proposed to be incorporated to DiT to improve the generalizability accordingly. Experimental results also demonstrated the enhanced performance from the incorporated local attention windows.

**Strengths:**

1. *Quality*: This work demonstrates good quality on identifying the inductive bias of DiT on generalizability and constructing the local attention map. The preliminary analysis and the model are of coherence and comprehensiveness.

2. *Significance*: Diffusion models have been attracting increasing attention in the recent years. Analysis the generalizability of the prominent diffusion models DiT via inductive biases provides an interesting perspective to improve diffusion models.

**Weaknesses:**

1. *Presentation*: A minor comment is on the grammar and spelling check of the paper, for example, in Section 1, “Their training involves approximates a distribution…” should be “Their training involves approximating a distribution…”.

2. *Clarity*: What does the Jacobian of a UNet or a DiT mean? Could the authors please articulate how it is computed as in [1] within the main content, for example, Section 2? Hence we can have a better understanding on why the Jacobian eigenvectors can reveal the memorization behavior of Simplified UNets, UNets and DiT.

3. *Novelty*: Although this work is of significance, the analysis perspective and method is still similar to [1]. It would be better to highlight the novelty and contributions of this work compared with [1], especially the differences on the analysis method and techniques. Theoretical analysis will be helpful to improve the novelty, as well as the quality.

[1] Zahra Kadkhodaie, Florentin Guth, Eero P Simoncelli, and Stéphane Mallat. Generalization in diffusion models arises from geometry-adaptive harmonic representation. In *ICLR*, 2024. 1, 2, 3, 4, 5, 10

**Questions:**

1. Could the experimental results also include the simplified UNet and UNets in improved diffusion models? Hence the comparison can be more comprehensive and promising.

2. It would also enhance the qualitative studies in the experimental results to show the improved generalizability of DiT with local attention windows. The impact of the local attention window size can also be presented.

---

> ### Author Response · Authors · 2024-11-23
>
> * **Q: Grammar and spelling.**
>
>   **A**: Thanks for letting us know. We uploaded a revised paper which corrects those typos.
> ---
>
> * **Q: The computation of the Jacobian eigenvector and why it can reveal the memorization behavior of Simplified UNets, UNets and DiT.**
>
>   **A**: The details of the Jacobian eigenvector computation have been moved from the appendix to Sec. 2 of the main paper (line 237-247). Specifically, we first feed a noisy image into a DiT or a UNet and obtain their Jacobian matrix $\nabla\epsilon_\theta$, where each entry of the Jacobian represents the partial derivative of an output pixel *w.r.t.* an input pixel. Then we perform the eigendecomposition of the Jacobian and obtain the eigenvectors and eigenvalues.
> ---
>
> * **Q: Novelty and contributions of this work compared with [1] regarding analysis method and techniques.**
>
>   **A**: The paper by Kadkhodaie et al. [1] reveals that harmonic bases are the inductive bias of **simplified UNets** that drive its generalization, measured by the PSNR gap. We do use their analysis method and find that a DiT does NOT exhibit this harmonic bases. Based on this finding, we introduce a new analysis of a DiT’s attention map and reveal the attention map locality as an inductive bias of a DiT. Our analysis of a DiT’s generalization and the conclusion differs from [1]. In addition, in Sec. 3, we verify the discovered inductive bias of a DiT by using local attention to modify a DiT’s generalization performance, which has not been explored by [1].
>
>   The theoretical analysis of the generalization of modern DiT and UNet architectures has been a challenge, mainly due to the complicated operations used by these networks. To make the theoretical analysis possible, [1] uses a one-channel simplified UNet, which is very different from normal widely used DiTs and UNets, leaving a gap between the theoretical analysis and networks we use in real world applications. This work focuses more on normal DiTs and UNets, providing insight that can be applied to modern diffusion models more directly, which we believe is equally important.
> ---
>
> * **Q: Including UNet in experiments for comprehensiveness.**
>
>   **A**: We included experimental results for the UNet in Tab. 2 (pixel-space DiT, PSNR Gap, line 324), Tab. 3 (pixel-space DiT, FID, line 335), and Tab. 4 (latent-space DiT, line 378) of the revised Appendix E.
>
>   | PSNR Gap (Pixel)  | $N{=}10^3$ | $N{=}10^4$ | $N{=}10^5$ |
>   | --- | --- | --- | --- |
>   | UNet (CelebA) | 13.86 | 5.53 | 0.06 |
>   | UNet (ImageNet) | 13.39 |  4.84 |  0.05 |
>   | UNet (MSCOCO) | 13.65 |  5.20 |  0.13 |
>
>   | FID (Pixel)  | $N{=}10^4$ | $N{=}10^5$ |
>   | --- | --- | --- |
>   | UNet (CelebA) | 9.8136 |  3.3871 |
>   | UNet (ImageNet) | 61.3965 |  13.1302 |
>   | UNet (MSCOCO) | 58.4580 |  7.0214 |
>
>   | PSNR Gap (Latent)  | $N{=}10^3$ | $N{=}10^4$ | $N{=}10^5$ |
>   | --- | --- | --- | --- |
>   | UNet (CelebA) | 8.69 |  3.35 |  0.11 |
>   | UNet (MSCOCO) | 5.74 |  1.84 |  0.12 |
>
>   | FID (Latent)  | $N{=}10^4$ | $N{=}10^5$ |
>   | --- | --- | --- |
>   | UNet (CelebA) | 36.5862 |  6.4805 |
>   | UNet (MSCOCO) | 159.7947 |  41.9908 |
>
> ---
>
> * **Q: Qualitative results.**
>
>   **A**: We included a qualitative comparison between DiT-XS/1 with and without local attention in Fig. 8 (line 472) of the revised Appendix F. For some samples, we find that DiTs trained with $10^4$ and $10^5$ images while using local attention produce images that are more like each other than DiTs trained without using local attention. We compute the averaged pixel intensity difference between DiTs trained with $10^4$ and $10^5$ images in Tab. 5 (line 415) of the revised Appendix F, which quantitatively verifies the above observation:
>
>   | Avg Pixel Diff | LSUN Bridge | LSUN Church | ImageNet|
>   | --- | --- | --- | --- |
>   | DiT-XS/1|  13.0078|  12.9620|  17.5844|
>   | DiT-XS/1 w/ Local | 11.0645|  10.9402 | 15.2034|
> ---
> [1] *Zahra Kadkhodaie, Florentin Guth, Eero P Simoncelli, and Stéphane Mallat. Generalization in diffusion models arises from geometry-adaptive harmonic representation. In ICLR, 2024*

---

> > ### Comment · Reviewer_oYGR · 2024-11-25
> >
> > Thank you for your response and explanations. Could you include the articulations on why the Jacobian eigenvectors can reveal the memorization behavior of Simplified UNets, UNets and DiT via your own explanations other than mathematical formula? In terms of the qualitative evaluations, would you present the generated images?
> >
> > Thanks a lot.

---

> > > ### Author Response · Authors · 2024-11-26
> > >
> > > Thank you very much for the new constructive comments. We address the new issues below:
> > >
> > > * **Q: Intuitive explanation why the Jacobian eigenvectors can reveal the memorization behavior of Simplified UNets, UNets and DiT.**
> > >
> > >   **A**: The Jacobian matrix represents for each output pixel the gradient *w.r.t.* all input pixels.  The eigendecomposition of the Jacobian matrix extracts the “principal components” and their corresponding eigenvalues. Note,
> > >   * Each “principal component” represent a **direction** which is used to compose the image generation; and
> > >   * The “principal component” corresponding to a larger eigenvalue is a more dominant direction.
> > >
> > >   For a simplified UNet and a UNet, when trained with $N{=}10$ images, their first few eigenvalues are significantly larger than all other eigenvalues, as illustrated in Fig. 3. This means that **1) the model’s output is dominated by a few “principal components” having large eigenvalues.** Meanwhile, **2) these dominant “principal components” dictate the composition of the training image.** Taking both points 1) and 2) into account means that only very few distinct image generations are possible. This indicates that the model is mostly generating memorized images rather than generalizing.
> > > ---
> > >
> > > * **Q: Regarding the qualitative evaluation images.**
> > >
> > >   **A**: We included a qualitative comparison between DiT-XS/1 with and without local attention in `Fig. 8 (line 472)` of the `revised Appendix F`.

---

> > > > ### Comment · Reviewer_oYGR · 2024-11-26
> > > >
> > > > Thank you for your reply. All my questions are now addressed and I would increase the score by 1.

---

> > > > > ### Author Response · Authors · 2024-11-26
> > > > >
> > > > > Thank you very much for your constructive comments which helped to clarify the presentation and made the experimental verification more comprehensive. We really appreciate your support of our paper and the novel insights for generative modeling that are discussed.

---

### Official Review · Reviewer_VbJL · 2024-11-03

**Soundness:** 2
**Presentation:** 3
**Contribution:** 2
**Rating:** 3
**Confidence:** 3

**Summary:**

This paper investigates the inductive bias of diffusion transformers. Specifically, the authors start by looking at the difference in the generalization behavior between UNet and DiT. Visualizing the eigenvectors of the harmonic bases, they show that the eigenvectors of smaller eigenvalues of UNet display more interesting patterns than DiT, while the PSNR gap between the training and testing dataset is large for UNet when the number of training data is smaller. Motivated by this difference, they visualize the attention maps of DiT with different training sizes and find out that the attention maps exhibited a more obvious locality pattern when the training size increased. At last, they propose to restrain the attention window size to enhance this locality property of the attention maps during the training. Their results show considerable improvement in the FID score when the training size is smaller.

**Strengths:**

The paper is well-presented and easy to follow. The authors conduct extensive experiments and show relative improvements in the image generalization quality on CelebA and ImageNet datasets.

Overall, I found the topic of looking into the generalization behavior of diffusion transformers very interesting.

**Weaknesses:**

1. The focus in the 1st part of the paper seems to disconnect from the rest, especially the method and the experimental results. I don't understand how the proposed method relates to the difference between the patterns of the harmonic bases for UNet and DiT. I am wondering do the authors observe differences in the harmonic bases after they alter the window size of the transformers.

2. The results are concerning. Though the current results show a considerable amount of improvement with a very limited size of the training data, the downgraded performance when scaling to a larger dataset makes it hard to believe that locality is the right reason behind the good generalization capability.

Typos and the minors:

Line 190: $\sigma_t$ is undefined.

Line 202: I assume you are talking about PSNR instead of PNSR, right?

**Questions:**

1. Eqn. 4: What does $\hat{x}_0^k$ mean? And what’s the intuition behind setting $K=300$?

2. What is the training time for different training sizes (i.e., N)? What's the norm of the attention maps when you increase the training time? Should this be considered as an impact on the emergence of the locality?

3. If you conduct the experiments in the latent space, have you observed similar results? Related to this, if you conduct the results on a more complex dataset where long-range correlations between objects matter(e.g., COCO-2017), what's your observation?

4. For Figure 4, can you show the results with different thresholds of the colormaps?

---

> ### Author Response · Authors · 2024-11-23
>
> * **Q: Connection between the harmonic bases analysis and the rest of the paper.**
>
>   **A**: This paper studies the inductive bias of diffusion transformers rather than proposing a method to improve a DiT. The first part of the paper shows that harmonic bases are **NOT** the inductive bias of a DiT, while the remaining paper demonstrates that attention locality **IS**. As outlined in the abstract, the previous study in [1] reveals harmonic bases as the inductive bias of a UNet that drives its generalization. For the inductive bias of a DiT, it is natural to speculate **whether a DiT also exhibits harmonic bases as its inductive bias?** Further study reveals that this is not the case. Instead, in Fig. 4 of the main paper, we find that locality of attention maps are the inductive bias that drives a DiT’s generalization. We then **verify** our findings by injecting local attention to a DiT and monitor the change of its generalization behavior, measured by PSNR gap and FID.
>
> ---
> * **Q: Differences in the harmonic bases when altering the attention window size.**
>
>   **A**: We add Fig. 4 to the revised supplementary material (line 232), which compares the Jacobian eigenvectors between DiT-XS/1 with and without using local attention. Their Jacobian eigenvectors show very similar sparse patterns and do not demonstrate harmonic bases observed in UNet.
>
> ---
> * **Q: Typos and the undefined notations**
>
>   **A**: Thanks for highlighting, we uploaded a revised paper.
>
> ---
> * **Q: What does $\hat{x}_k$ mean in Eq. 4? And what’s the intuition behind setting $K{=}300$?**
>
>   **A**: $\hat{x}^k_0$ denotes the estimated $x_0$ for image $k$ at diffusion step $t$ using Eq. (3). We added the definition of $\hat{x}^k_0$ in the revised paper (line 194). We set $K{=}300$ following [1]. Moreover, Fig. 2 of the main paper shows that PSNRs for different images have a very low variance, suggesting that $K{=}300$ is enough to obtain reliable PSNRs of a model.
>
> ---
> * **Q: Training time for different N, the norm of attention maps, and their impact on the emergence of locality.**
>
>   **A**: As stated in line 372, we train a DiT with fixed $400k$ iterations for different training images $N$. All other hyper-parameters including the **norm of attention maps** are kept the same for all experiments and visualizations. We believe both training time and norm of attention maps are not impacting the emergence of the attention map locality.

---

> ### Author Response · Authors · 2024-11-23
>
> * **Q: Experimental results on MSCOCO 2017 dataset and using latent-space diffusion model.**
>
>   **A**: We include experimental results on MSCOCO data in Tab. 2 (PSNR gap, line 324) and Tab. 3 (FID, line 335) of the revised Appendix E.1.
>
>   For the PSNR gap comparison using MSCOCO data we obtain:
>
>   | PSNR Gap (MSCOCO)  | $N{=}10^3$ | $N{=}10^4$ | $N{=}10^5$ |
>   | --- | --- | --- | --- |
>   | DiT-XS/1  | 7.36 | 0.60 | 0.13 |
>   | DiT-XS/1 w/ Local | 6.36 | 0.41 | 0.13 |
>
>   Using local attention can reduce the PSNR gap when $N{=}10^3$ and $N{=}10^4$, which aligns well with our observations on other datasets shown in Tab. 1 of the main paper.
>   For the FID comparison using MSCOCO dataset we obtain:
>
>   | FID (MSCOCO)  | $N{=}10^4$ | $N{=}10^5$ |
>   | --- | --- | --- |
>   | DiT-XS/1  | 28.3496 | 12.9695 |
>   | DiT-XS/1 w/ Local | 24.4308 | 13.4735 |
>
>   Using local attention reduces the FID when the number of training images is small, which also aligns with the results on other datasets as shown in Fig. 6 of the main paper.
>
>   In addition, we include experimental results based on a latent diffusion model in Tab. 4 (line 378) of the revised Appendix E.2, using CelebA and MSCOCO data:
>
>   | PSNR Gap (CelebA)  | $N{=}10^3$ | $N{=}10^4$ | $N{=}10^5$ |
>   | --- | --- | --- | --- |
>   | DiT-XS/1  | 3.36 | 1.19 | 0.07 |
>   | DiT-XS/1 w/ Local | 2.21 | 0.17 | 0.07 |
>
>   | FID (CelebA)  | $N{=}10^4$ | $N{=}10^5$ |
>   | --- | --- | --- |
>   | DiT-XS/1  | 12.7045 | 8.5751 |
>   | DiT-XS/1 w/ Local | 13.6513 | 9.2621 |
>
>   | PSNR Gap (MSCOCO)  | $N{=}10^3$ | $N{=}10^4$ | $N{=}10^5$ |
>   | --- | --- | --- | --- |
>   | DiT-XS/1  | 2.08 | 0.14 | 0.11 |
>   | DiT-XS/1 w/ Local | 1.19 | 0.13 | 0.10 |
>
>   | FID (MSCOCO)  | $N{=}10^4$ | $N{=}10^5$ |
>   | --- | --- | --- |
>   | DiT-XS/1  | 72.3063 | 68.1251 |
>   | DiT-XS/1 w/ Local | 78.8400 | 74.1681 |
>
>   Comparing DiT-XS/1 with and without local attention, we observe the use of local attention to reduce the PSNR gap. However, we do not observe a smaller FID value when $N$ is small, making it different from the pixel-space DiT. To investigate this further, we compare the attention map between pixel-space and latent-space DiTs in Fig. 7 of the revised appendix. We observe that the attention map locality gap between $N{=}10^3$ and $N{=}10^5$ is larger in a pixel-space DiT than in a latent-space DiT. We speculate that this is because larger training images ($256{\times}256$ for the latent DiT compared with $32{\times}32$ for pixel DiT), coupled with the VAE encoder, create more diverse information that enables a latent DiT to more easily achieve good generalization (reflected by attention map locality). Because of this, it is hard to improve a DiT's FID further when $N$ is small.
>
> ---
> * **Q: For Figure 4, can you show the results with different thresholds of the colormaps?**
>
>   **A**: We add new DiT attention map visualization results in Fig. 5 (line 284) and Fig. 6 (line 302) of the revised Appendix D, using threshold 0.3 and 0.5, respectively. Both figures show the same locality trend when increasing the number of training images $N$.
>
> ---
> [1] *Zahra Kadkhodaie, Florentin Guth, Eero P Simoncelli, and Stéphane Mallat. Generalization in diffusion models arises from geometry-adaptive harmonic representation. In ICLR, 2024*

---

> > ### Author Response · Authors · 2024-11-26
> >
> > Thank you for your valuable comments that help us improve this paper. We hope our response as well as the revised paper and supplementary material are able to answer your questions. We are looking forward to hearing back from you and answering any new questions that you may have. Thank you again for your effort to improve this paper!

---

> > ### Comment · Reviewer_VbJL · 2024-12-01
> >
> > I want to first thank the authors for the detailed reply and the new experimental results. I understand the paper focuses on analyzing the DIT's generalization behavior instead of proposing a new method. However, my biggest concern still holds that the authors do not provide sufficient evidence for the claim that attention locality is what impacts the generalization behavior of DIT, especially when the current results show that the image quality decreases when adding local window constraint as the number of data samples increases (e.g., N=10^5). Since we know that the practical dataset is considerably large (e.g., the entire ImageNet-1K contains O(10^7) images), I do feel the authors need to provide evidence on large-scale datasets to support their claim.

---

> > > ### Author Response · Authors · 2024-12-02
> > >
> > > Thanks for participating in the discussion and for valuable feedback. We think there might be a misunderstanding regarding model generalization (opposite of overfitting) and data scaling. To address this possible misunderstanding, we first clarify the difference between model generalization and scaling. Then we present new experimental results training with $10^6$ images on ImageNet (*i.e.*, the entire ImageNet-1k dataset) and LSUN Bedroom data.
> > >
> > > * **Generalization $\neq$ Data Scaling.**
> > >
> > >   The model generalization studied in this paper differs from data scaling (*e.g.*, scaling laws) of a model. We study generalization of a model, *i.e.*, the model’s performance gap between the training and testing dataset. To assess the performance gap, we use the PSNR gap between the training and testing sets following prior work [1].
> > >
> > >   Our paper finds “attention locality to impact the generalization behavior of a DiT”, *i.e.*, attention locality impacts the PSNR gap between the training and testing sets. As shown in Fig. 6, when the training dataset size is small (*e.g.*, $N{=}10^3$ or $N{=}10^4$), using local attention prevents a DiT from overfitting on the training set and reduces the DiT PSNR gap. When the training dataset size is sufficient (*e.g.*, $N{=}10^5$), using local attention has little impact on the PSNR gap.
> > >
> > >   Note, FID and DiT’s generation quality when scaling the training set are not the evidence we use to verify our claim. In contrast, as stated in line 399-400, we solely present FID results in order to provide insights about how the FID will change if a DiT’s generalization is modified. Importantly, the FID trend of a DiT when scaling training set size is not related to the contribution of this paper.
> > >
> > >   We will include this clarification in the revised paper to reduce the risk for a misunderstanding. We are happy to provide additional clarification if the reviewer feels this statement is not clear enough.
> > >
> > > ---
> > > * **Why training a DiT with $10^5$ images is enough to verify our claim.**
> > >
> > >   Our initial experiments use at most $10^5$ images for the following two reasons:
> > >   * Prior work [1] studies the generalization of a simplified UNet. We follow their settings and use at most $10^5$ images.
> > >   * More importantly, as shown in Fig. 6, a training set of $10^5$ images already results in a very low PSNR gap, *i.e.*, a DiT’s generalization is almost saturated when $N{=}10^5$. Improving the generalization of a DiT trained with $N{=}10^5$ or more images is unnecessary.
> > >
> > > ---
> > > * **More experimental results with $10^6$ images on ImageNet and LSUN Bedroom data.**
> > >
> > >   Following the reviewer’s comments, we conduct new experiments, training DiT-XS/1 with and without local attention using $10^6$ images on ImageNet and LSUN Bedroom data. We use the entire ImageNet (1,281,167 images) dataset as 281,167 images are used for testing. Below we show the PSNR and FID comparison between DiT-XS/1 with and without local attention.
> > >
> > >   | PSNR Gap (ImageNet)   | $N{=}10^3$ | $N{=}10^4$ | $N{=}10^5$ | $N{=}10^6$ |
> > >   | --- | --- | --- | --- | --- |
> > >   | DiT-XS/1 | 7.77 |  1.08 |  0.05  | 0.04 |
> > >   | DiT-XS/1 w/ Local  | 6.76 |  0.74 |  0.05 | 0.04 |
> > >
> > >   | FID (ImageNet)  | $N{=}10^4$ | $N{=}10^5$   | $N{=}10^6$ |
> > >   | --- | --- | --- | --- |
> > >   | DiT-XS/1 | 36.8461 | 20.1907 | 19.2973 |
> > >   | DiT-XS/1 w/ Local | 31.4555 | 20.3175 | 19.9701 |
> > >
> > >   | PSNR Gap (LSUN Bedroom)  | $N{=}10^3$ | $N{=}10^4$ | $N{=}10^5$   | $N{=}10^6$ |
> > >   | --- | --- | --- | --- | --- |
> > >   | DiT-XS/1 | 7.45 | 0.70 | 0.26  | 0.24 |
> > >   | DiT-XS/1 w/ Local | 6.20 | 0.52 | 0.26 | 0.24 |
> > >
> > >   | FID (LSUN Bedroom)  | $N{=}10^4$ | $N{=}10^5$  | $N{=}10^6$ |
> > >   | --- | --- | --- | --- |
> > >   | DiT-XS/1 | 15.6740 | 4.8256  | 4.5789 |
> > >   | DiT-XS/1 w/ Local | 11.2033 | 5.0868 | 5.0227 |
> > >
> > >   For both datasets, the PSNR gap of a DiT-XS/1 trained with $10^6$ images is already very small, *i.e.*, a DiT-XS/1 generalizes when trained with $10^6$ images. Using local attention has limited impact for models which already generalize. Regarding FID, using local attention slightly increases the FID on both datasets. Again as discussed above, FID of a DiT trained with $N{=}10^6$ is unrelated to our main claim that attention locality impacts the generalization behavior of a DiT.
> > > ---
> > > [1] *Zahra Kadkhodaie, Florentin Guth, Eero P Simoncelli, and Stéphane Mallat. Generalization in diffusion models arises from geometry-adaptive harmonic representation. In ICLR, 2024*

---

> > > > ### Author Response · Authors · 2024-12-02
> > > >
> > > > Thank you for your active participation in the discussion process. Since the discussion period ends today, we would like to kindly ask whether we have addressed all of your questions regarding the verification of our claim that attention locality is the inductive bias that enables the generalization of a DiT. We are happy to address any remaining questions if any. Thank you again and we are looking forward to your reply.

---

### Author Response · Authors · 2024-11-23
**General response**

We thank all reviewers for their valuable comments. We are encouraged that reviewers recognize the following merits of this paper:
* The studied topic is interesting, significant, and highly-relevant (Reviewer `VbJL`, `oYGR`, `pa74`, `MCon`)
* The paper is of high-quality and well-presented (Reviewer `VbJL`, `oYGR`)
* The analysis is coherent, comprehensive, and interesting (Reviewer `oYGR`, `pa74`)
* The experimental verification is extensive and sufficient (Reviewer `VbJL`, `pa74`)

We address the raised questions one-by-one below. Please note we also revised the paper to include new experimental results and visualization in response to reviewers’ questions and comments. Newly added and modified text, figures, and tables are highlighted with **red** color.

---

### Author Response · Authors · 2024-12-04
**Author Response Summary**

Dear Reviewers, ACs, SACs, PCs:

Thanks a lot for your valuable comments and active participation in the discussion. We think that all of the questions raised by reviewers have been thoroughly answered. Below we restate our contribution before summarizing how we addressed the reviewers’ questions.

This paper studies the inductive biases of diffusion transformers (DiTs) that enable their generalization. Through an empirical analysis, we identify the locality of attention maps as a key inductive bias, which contributes to the generalization of a DiT. To verify this inductive bias, we present experimental results by incorporating local attention into a DiT. We show that enhancing the locality of attention can effectively modify a DiT’s generalization. We added a theoretical explanation showing that the attention locality bias is connected with the low sensitivity bias studied by prior work.

During the discussion period, we address the questions of all reviewers as summarized below:
* Reviewer `VbJL`:
  * **Differences in the harmonic bases when altering the attention window size**: We added Fig. 4 to the revised Appendix F. It shows the harmonic bases after applying local attention.
  * **New experiments with MSCOCO and latent space DiT**: We analyzed and presented the results on MSCOCO in Tabs. 2 and 3 of the revised Appendix E.1. We included experimental results for a latent diffusion model in Tab. 4 of the revised Appendix E.2, using CelebA and MSCOCO data.
  * **Possible misunderstanding of the term “generalization”**: We provided an explanation to note that the generalization we study is not identical to the data scaling capability of a DiT.
  * **New experiments with a training set of $10^6$ images**: We provided results with $10^6$ images using ImageNet and LSUN Bedroom data. We explained why we think using $10^5$ images is sufficient to verify the discovered inductive bias. Results with $10^6$ images corroborate this.
* Reviewer `oYGR`:
  * **Details about the Jacobian eigenvector computation**: We moved details about the Jacobian eigenvector computation from the Appendix to Sec. 2 of the paper.
  * **New experiments with UNet**: We included experimental results for the UNet in Tabs. 2, 3, and 4 of the revised Appendix E.
  * **Qualitative results**: We provided qualitative results in Fig. 8 of the revised Appendix F.
* Reviewer `pa74`:
  * **Details about the Jacobian eigenvector computation**: We provided an explanation in the response and moved the details about the Jacobian eigenvector computation from the Appendix to Sec. 2 of the paper.
  * **Details about token dimension**: We provided an explanation in the response.
* Reviewer `MCon`:
  * **Effectiveness of local attention compared with simply reducing model complexity**: We provided an explanation noting that reducing the model complexity does not always lead to the improved generalization. We added new experiments with a smaller DiT-XXS/1 in Tabs. 2 and 3 of Appendix E, showing that reducing model complexity is not as effective as local attention.
  * **The contribution of this paper considering that local attention has been widely used**: We provided an explanation stating that our goal isn’t to improve a DiT’s performance via local attention. In contrast, we recognize attention locality as a key inductive bias of DiT. Local attention serves as a way to encourage attention locality of a DiT. We solely apply local attention to verify our discovered inductive bias rather than as a contribution of this work.
  * **New experiments with MSCOCO**: We included experimental results using MSCOCO data in Tabs. 2 and 3 of the revised Appendix E.1.
  * **Qualitative results**: We provided qualitative results in Fig. 8 of the revised Appendix F.
  * **Theoretical explanation for why local attention improves generalization of a DiT**: We provided a theoretical explanation in Appendix H linking the attention locality bias with the low sensitivity bias, flat minima, and generalization. We also provided a line-by-line explanation of the theoretical derivation.

In summary, we think we have comprehensively addressed all reviewers' comments. We are optimistic that our work provides valuable insights for generative modeling, facilitating understanding of the behavior of a DiT. Lastly, we thank everyone once again for their time and feedback, which helped us to improve the paper.

---

### Meta-Review · Area_Chair_kTL3 · 2024-12-12

**Metareview:**

This paper explores the generalization behaviour of Diffusion Transformers, and claims that attention locality is the inductive bias, as opposed to UNet-based diffusion models which were said to have harmonic bases as inductive bias in prior work. Local attention windows are proposed as a way to improve generalizability.

However, reviewers felt that the experimental evaluation was not sufficient to robustly back up these claims. In particular, there can be alternative explanations for the beneficial effect of local attention windows on small datasets which were not ruled out by the authors. For example a more mundane explanation that they simply decrease the model complexity improving generalizability, rather than having a beneficial impact on the inductive bias. The authors have failed to consistently apply the scientific method by making claims without sufficient evidence and considering alternatives, and hence I am recommending rejection.

**Additional Comments On Reviewer Discussion:**

The main points of improvement noted by reviewers were: the paper was initially not self-contained and needed clarification especially around the use of Jacobians; the experimental results do not sufficiently support the conclusions; there was initially a lack of theoretical analysis; generally the work is similar to a past work, and the authors should more clearly justify the novelty of their contributions.

Improvements were made to the description of Jacobians. Theoretical analysis was added, but only during discussion and therefore may not have been reviewed at the standard required by ICLR.

Reviewers also noted a heavy reliance on FID scores, and that FID is widely criticized in the literature. While the reviewers recommended alternatives like CLIP scores, the authors noted that CLIP score is not appropriate for their setting. However, the weaknesses of FID are still relevant, especially when working with diffusion models (e.g. Stein et al. “Exposing flaws of generative model evaluation metrics and their unfair treatment of diffusion models” NeurIPS 2023), so the authors should consider alternative measures of performance.

---

### Decision · Program_Chairs · 2025-01-22

Reject